# GENERATIVE LEARNING FOR FINANCIAL TIME SERIES WITH IRREGULAR AND SCALE-INVARIANT PATTERNS

## IN MEMORY OF PROF. DUAN LI

**Hongbin Huang, Minghua Chen**[*]**, and Xiao Qiao**[*]

School of Data Science, City University of Hong Kong

`hongbin.huang@my.cityu.edu.hk,{minghua.chen,xiaoqiao}@cityu.edu.hk`

## ABSTRACT

Limited data availability poses a major obstacle in training deep learning models for financial applications. Synthesizing financial time series to augment real-world data is challenging due to the *irregular and scale-invariant* patterns uniquely associated with financial time series - temporal dynamics that repeat with varying duration and magnitude. Such dynamics cannot be captured by existing approaches which often assume regularity and uniformity in the underlying data. We develop a novel generative framework called *FTS-Diffusion* that consists of three modules to model irregular and scale-invariant patterns. First, we present a scale-invariant pattern recognition algorithm to extract recurring patterns that vary in duration and magnitude. Second, we construct a diffusion-based generative network to synthesize segments of patterns. Third, we model the temporal evolution of patterns in order to aggregate the generated segments. Extensive experiments show that FTS-Diffusion generates synthetic financial time series highly resembling observed data, outperforming state-of-the-art alternatives. Two downstream experiments demonstrate that augmenting real-world data with synthetic data generated by FTS-Diffusion reduces the error of stock market prediction by up to 17.9%. To the best of our knowledge, this is the first work on generating intricate time series with irregular and scale-invariant patterns, addressing data limitation issues in finance.

## 1 INTRODUCTION

Researchers in financial economics have demonstrated intriguing potential for deep learning to solve complex problems in financial settings (Qin et al., 2017; Xu & Cohen, 2018; Wu et al., 2020; Manzo & Qiao, 2020; Huang & Li, 2021). However, a dearth of data and the low signal-to-noise ratio nature of financial data pose major obstacles that hinder the further development of deep learning in finance. Unlike the sciences, finance researchers cannot run experiments to obtain more data, so financial time series are limited by their existing history. Additionally, price and return data are subject to high levels of noise, making it even more challenging to extract useful information from a limited dataset. Deep learning models trained on insufficient data are prone to overfitting and cannot be expected to perform reliably on unseen data.

To alleviate data scarcity, data augmentation techniques can be employed. Generative models that capture the properties of the underlying data-generating process would produce synthetic data that resemble observed data. Recently, deep generative modeling, especially generative adversarial networks (GAN) (Goodfellow et al., 2014) and diffusion models (Ho et al., 2020), has made remarkable progress in multiple domains including image synthesis, reinforcement learning, and anomaly detection[1]. They have also been applied to time series settings such as medical records, audio synthesis, power systems, and networked systems[2]. Despite these advances, modeling financial time series poses unique challenges that complicate the task and render existing models ineffective.

---

[*]Corresponding authors: Minghua Chen and Xiao Qiao.

[1]Papers on images include Zhu et al. (2017), Turkoglu et al. (2019), Dhariwal & Nichol (2021), and Rombach et al. (2022). Reinforcement learning works include Yu et al. (2017), Janner et al. (2022), and Eysenbach et al. (2022). Anomaly detection studies include Zenati et al. (2018) and Akcay et al. (2019).

[2]Medical: Esteban et al. (2017), Che et al. (2017), and etc.Audio: Kong et al. (2021), Leng et al. (2022), and etc. Power systems: Zhang et al. (2018) and Chen et al. (2018). Networked systems: Lin et al. (2020).

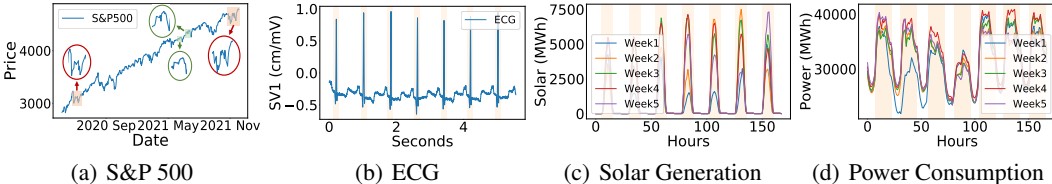

(a) S&P 500       (b) ECG       (c) Solar Generation       (d) Power Consumption

Figure 1: Time-series data of (a) S&P 500 price (finance), (b) ECG (medical), (c) solar generation (renewable energy), and (d) power consumption (smart grid). Unlike other time-series data with explicit patterns, financial time series expresses complex patterns that are irregular and scale-invariant.

The time series studied in the extant literature of deep generative learning tend to exhibit some regularity. Patterns identified in these data appear at fixed or predictable increments in calendar time (e.g., heartbeats in ECG). Time series data that contain such *regular* patterns are amenable to modeling, as they allow the extraction of highly correlated features from similar repeating patterns. Although conceptually straightforward, identifying recurring patterns in financial time series proves difficult due to a lack of regularity. Instead, financial time series appear to contain more subtle patterns that repeat themselves with varying duration and magnitude, a quality we refer to as *scale-invariance*. *Irregularity* and *scale-invariance* are hallmarks of financial time series that complicate their modeling and the synthesis of additional data. We illustrate these two properties in Fig. 1 by comparing the S&P 500 Index, a broad basket of U.S. stocks, to several regular series. The three regular series exhibit clear and consistent patterns that align with calendar time. In contrast, we do not observe neat patterns that adhere to a fixed frequency for the S&P 500. Instead, we can observe similar patterns (red and green circles) that exhibit scale-invariance. These patterns keep their basic shape but are shifted or stretched compared to each other[3]. The unique properties of financial time series make data synthesis a significantly more challenging task compared to that of well-behaved data. Effective time series data generation considering irregularity and scale-invariance remains largely an open problem.

To address this problem, we deconstruct financial time series generation into a three-prong process: (i) *pattern recognition* to identify irregular and scale-invariant patterns, (ii) *generation* to synthesize segments of patterns, and (iii) *evolution* to connect the generated segments into a complete time series. We propose a new generative framework, *FTS-Diffusion* to accomplish the *pattern recognition-generation-evolution* process. We find that FTS-Diffusion is capable of generating synthetic financial time series that closely resemble observed data. We make the following contributions:

▷ We identify and define two properties of financial time series: irregularity and scale-invariance (see Sec. 3). We present a novel FTS-Diffusion framework to model time series data exhibiting these properties. To the best of our knowledge, this is the first framework capable of generating challenging time series data that contain irregularity and scale-invariance. FTS-Diffusion may also be applied to other domains with data exhibiting similar properties.

▷ The unique architecture of FTS-Diffusion is designed to handle irregularity and scale-invariance. There are three modules. The pattern recognition module is based on a new scale-invariant subsequence clustering (SISC) algorithm (Sec. 4.1). By incorporating dynamic time warping (DTW), SISC is able to accurately identify and separate irregular and scale-invariance patterns. The generation module consists of a diffusion-based network to synthesize the scale-invariant segments conditional on the patterns learned by SISC (Sec. 4.2). The evolution module is made up of a pattern transition network that produces the temporal evolution of consecutive patterns, capturing the dynamic relationship among them (Sec. 4.3).

▷ We demonstrate the effectiveness of FTS-Diffusion in capturing real-world financial data and we illustrate the value of the generated data for downstream applications (Sec. 5). Patterns identified by FTS-Diffusion can be cross-verified with financial domain knowledge (Lo et al., 2000), and experimental results from three real-world datasets show that FTS-Diffusion generates the most realistic financial time series among several alternative models. We explore the usage of the generated data for the downstream task of predicting stock prices. Augmenting limited real-world data with synthetic samples from FTS-Diffusion reduces the predictive error by up to 17.9% across the datasets.

---

[3]For more details, Appendix A provides an in-depth discussion on the properties of financial time series.

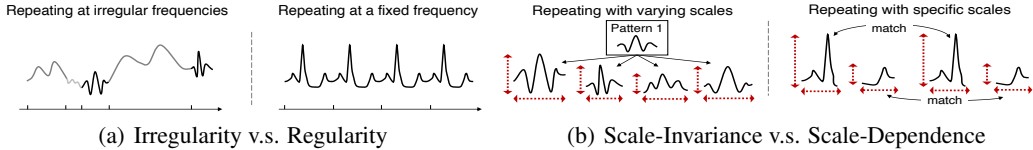

| (a) Irregularity v.s. Regularity | (b) Scale-Invariance v.s. Scale-Dependence |
| --- | --- |

Figure 2: (a) Irregular patterns with indeterminate recurrence intervals vs. regular patterns repeated at a fixed frequency. (b) Scale-invariant patterns showing similarity after proper duration/magnitude scaling vs. scale-dependent patterns showing similarity given specific duration or magnitude.

These results shed light on the capability of FTS-Diffusion to improve the accuracy and reliability of deep learning models in financial applications. Codes are available in supplementary materials.

## 2  RELATED WORK

Advances in deep generative modeling have shown promise to generate time series data in various problem domains, particularly using VAEs-, GANs-, and diffusion-based models. We discuss the most relevant works in this section.

TimeVAE (Desai et al., 2021) presents a VAE-based framework to model the trend and seasonality in time series. RCGAN (Esteban et al., 2017) and MV-GAN (Brophy, 2020) use GANs for learning medical records. Several GAN variants are employed to model time series in power systems (Zhang et al., 2018; Chen et al., 2018). TimeGAN (Yoon et al., 2019) develops a general framework to embed time-series data into a latent space with an autoencoder network and subsequently learn the latent representation with GANs. QuantGAN (Wiese et al., 2020) provides a GAN-based network to capture long-range dependencies in financial time series under the volatility-innovation decomposition. CSDI (Tashiro et al., 2021) proposes a score-based diffusion model primarily designed for imputation, with an unconditional variant that can be also used for time series generation. DiffWave (Kong et al., 2021) and BinauralGrad (Leng et al., 2022) generate waveform time series with diffusion models.

The above approaches can model time series with regular patterns but struggle with more complex series characterized by irregularity and scale-invariance, central features in financial time series. The identification of latent patterns in financial time series is challenging, and it is difficult for a generative model without auxiliary information to distinguish between these diverse distributions. In our study, we decompose the financial time series generation into a *pattern recognition-generation-evolution* process, enabling better modeling of the irregular and scale-invariant properties. In addition, diffusion probabilistic models have been shown to achieve better quality and training stability than the classical GAN and VAE models (Dhariwal & Nichol, 2021; Wang et al., 2021). Hence, we design our generative model leveraging the denoising diffusion probabilistic model (DDPM) (Ho et al., 2020).

## 3  PROBLEM STATEMENT

### 3.1  UNIQUE CHARACTERISTICS OF FINANCIAL TIME SERIES

The irregular and scale-invariant patterns in financial time series are difficult for existing models that assume regularity and uniformity to capture. Fig. 2 illustrates these repeating temporal dynamics with non-deterministic intervals and varying duration and magnitudes. The typical technique of dividing time series into fixed-interval segments in existing approaches is likely to result in a snapshot of either a fraction of a pattern or a mixture of multiple patterns.

We propose a novel framework to model irregular and scale-invariant time series. A time series $\boldsymbol{X} = \{\boldsymbol{x}_1, \ldots, \boldsymbol{x}_M\}$ consists of $M$ segments, $\boldsymbol{x}_m = \{x_{m,1}, \ldots, x_{m,t_m}\}$. The length of the entire time series is $T = \sum_{m=1}^{M} t_m$. $\boldsymbol{x}_m$ is sampled from a conditional distribution $f(\cdot | p, \alpha, \beta)$ dependent on the pattern $p \in \mathcal{P}$, whose duration is scaled by $\alpha$ and magnitude scaled by $\beta$. This way, $\boldsymbol{x}_m$ will be statistically similar to its underlying pattern $p$ while allowing for adjustments in duration and magnitude. To model the dynamics across patterns, we employ a Markov chain. Each tuple $(p, \alpha, \beta)$ is a state, and the state transition probabilities $Q(p_j, \alpha_j, \beta_j | p_i, \alpha_i, \beta_i)$ describe the stochastic

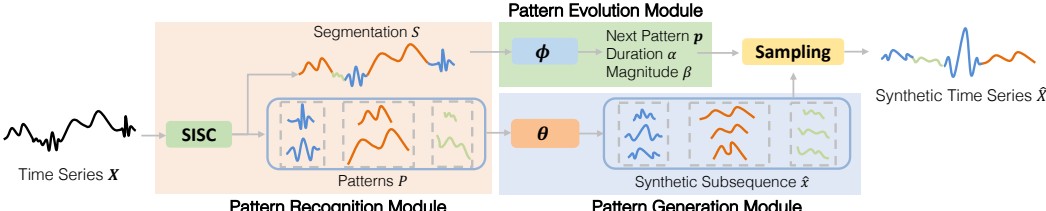

Figure 3: The architecture of our proposed FTS-Diffusion. The pattern recognition module identifies the scale-invariant patterns within the entire financial time series using our proposed SISC algorithm. Subsequently, the pattern generation module synthesizes the segments with a diffusion-based network conditioned on the patterns. Finally, the pattern evolution module connects the generated segments to construct a synthetic financial time series following the transition between consecutive patterns.

transition from one pattern to the next. Our setup is reminiscent of applications of the Markov property in financial time series (Dueker, 1997; Bai & Wang, 2011; Somani et al., 2014). The novelty in our approach is that we use a Markov model to capture the transition of three specific aspects of the time series: pattern, duration, and magnitude, whereas existing work attempts to recover some unspecified latent properties of a time series.

## 3.2 PROBLEM STATEMENT

We seek to operationalize the structure laid out in Sec. 3.1. When faced with a time series, we have no knowledge of the segments $\{\boldsymbol{x}_m\}_{m=1}^M$, the set of scale-invariant patterns $\mathcal{P}$, or the scaling factors $\alpha$ and $\beta$ that transform a reference pattern into its more realistic counterpart. We also do not know the transition probabilities $Q(p_j, \alpha_j, \beta_j | p_i, \alpha_i, \beta_i)$. Our goal is to develop a data-driven framework to accomplish the following:

- (Pattern Recognition) identify the patterns and learn the recurrent structures $\mathcal{P}$, and group segments into clusters according to their corresponding patterns $p \in \mathcal{P}$;
- (Pattern Generation) learn the distribution $f(\cdot | p, \alpha, \beta), \forall p \in \mathcal{P}$;
- (Pattern Evolution) learn the pattern transition probabilities $Q(p_j, \alpha_j, \beta_j | p_i, \alpha_i, \beta_i)$.

The three components above allow us to generate financial time series by (i) determining the allocation of patterns using the pattern transition probabilities, and (ii) generating each segment from the corresponding pattern with the appropriate duration and magnitude scaling factors. Our three-pronged framework dedicated to identifying and modeling the irregular and scale-invariant patterns observed in financial time series is the first of its kind in the literature.

## 4 OUR PROPOSED FTS-DIFFUSION FRAMEWORK

In this section, we present our FTS-Diffusion framework. Fig. 3 provides an illustration. FTS-Diffusion consists of three components: a pattern recognition module, a pattern generation module, and a pattern evolution module. Next, we introduce each module and how they work together.

### 4.1 PATTERN RECOGNITION: IDENTIFYING IRREGULAR AND SCALE-INVARIANT PATTERNS

We propose a novel Scale-Invariant Subsequence Clustering (SISC) algorithm to partition the entire financial time series into segments of variable lengths and group them into $K$ distinct clusters. The segments within the same cluster exhibit similar shapes after proper scaling in duration and magnitude. The centroid of each cluster then represents a scale-invariant pattern in the financial time series.

The idea is similar to the traditional K-Means clustering (Hartigan & Wong, 1979), which primarily clusters segments of identical length and thus falls short in our context due to its inability to handle

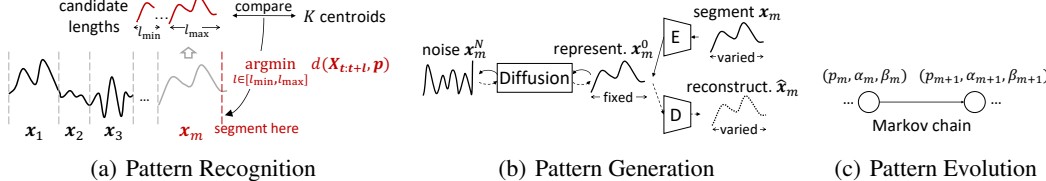

(a) Pattern Recognition      (b) Pattern Generation      (c) Pattern Evolution

Figure 4: Key designs in each module of our FTS-Diffusion: (a) pattern recognition module: SISC greedy segmentation; (b) pattern generation module: the pattern-conditioned diffusion network paired with the scaling autoencoder; (c) pattern evolution module: Markov transition.

segments of varying lengths and magnitudes. Instead of separating the entire time series into equal-length segments as is commonly done, we adaptively determine the optimal segment lengths through a simple yet effective greedy segmentation strategy. Specifically, as illustrated in Fig. 4(a), we compare the segments of candidate lengths $l \in [l_{\min}, l_{\max}]$ with the cluster centroids within a normalized space at each evaluated position $t = \sum_{\tau=0}^{m-1} t_\tau$. The length $l^*$ that minimizes the distance to the nearest centroid is considered the optimal segmentation for the current segment $\boldsymbol{x}_m = X_{t:t+l^*}$, i.e.,

$$l^* = \underset{l \in [l_{\min}, l_{\max}]}{\arg\min} \ d(X_{t:t+l}, \boldsymbol{p}), \ \forall \boldsymbol{p} \in \mathcal{P}. \tag{1}$$

**The first key component** in our design is a distance metric $d(\cdot, \cdot)$ that is robust to varying lengths and magnitudes and hence properly measures the difference between subsequences. Classical metrics, such as Euclidean distance, fail to provide accurate measurements due to their limitations in comparing variable-length sequences. In contrast, we employ dynamic time warping (DTW) to calculate the minimum distance across all pointwise alignments between two segments:

$$DTW(\boldsymbol{x}, \boldsymbol{y}) := \min_{A \in \mathcal{A}} \ \langle A, \Delta(\boldsymbol{x}, \boldsymbol{y}) \rangle, \tag{2}$$

where $A$ denotes the alignment between two sequences in the set of all possible alignments $\mathcal{A}$, and $\Delta(x, y) = [\delta(x_i, y_j)]_{ij}$ is the pointwise distance matrix between two normalized sequences $\boldsymbol{x}$ and $\boldsymbol{y}$. It well-suits our purpose to identify similarities in segments with similar shapes but varying duration and magnitudes. With the DTW metric as $d(\cdot, \cdot)$, we apply the greedy segmentation strategy from the start to the end of the time series. Upon completing the greedy segmentation across the entire time series, we proceed with the standard K-means clustering process. Each segment is assigned to the nearest centroid, then the centroids are updated based on current cluster assignments. This process iterates until cluster assignments stabilize or a pre-determined number of iterations is reached.

**The second key component** in our design is the initialization of the cluster centroids. The random initialization typically used in standard clustering methods often yields suboptimal results. To alleviate this issue, we design a wise initialization for more informed initial centroids. Our initialization begins by randomly selecting one segment from all available segments of a pre-specified length, which could be either the minimum or maximum length in practice, to serve as the first centroid. Afterward, we choose the subsequent centroids from the rest, with the selection weight being proportional to their distances to the closest centroid within the chosen set. This means that segments located farther from their nearest centroid have a higher probability of being the next choice. We repeat this process until $K$ centroids have been initialized. This method ensures a diverse set of centroids spreading across the data space, promoting an efficient start of our SISC algorithm.

We highlight that our SISC algorithm is the first in the literature designed to identify scale-invariant patterns. The computational complexity of SISC is $\mathcal{O}(TKl_{\max})$, which is linear to the length of the entire time series. The pseudo-code of SISC and the selection of parameters, such as the range of segment lengths and the number of clusters $K$, are detailed in Appendix B.1. The learned patterns are cross-validated with the technical patterns in the financial literature (Lo et al., 2000) in Appendix B.2. To better verify the effectiveness of our proposed SISC algorithm, we conduct a thorough investigation using simulated time series data in Appendix B.3.

## 4.2 PATTERN GENERATION: LEARNING PATTERN-CONDITIONED TEMPORAL DYNAMICS

We develop a pattern generation module, $\theta$ in Fig. 3, to synthesize the segments of patterns. The goal is to generate new segments that mimic the temporal dynamics within the observed segments. Considering the financial time series as a collection of scale-invariant patterns, the data-generating process can be interpreted as capturing the distribution of the reference patterns and transforming them with proper scales in duration and magnitude. Accordingly, we instantiate this data-generating process using two dedicated networks for the two tasks, as discussed below.

**The first network** is the scaling autoencoder (AE) for learning the transformation between variable-length segments $x$ and fixed-length representations $x^0$, after we capture the reference pattern representation using the pattern-conditioned diffusion network. The encoder of the scaling AE stretches the variable-length segments into fixed-length representations that align with the dimension of reference patterns. The decoder, on the other hand, is responsible for reconstructing the variable-length segments from the fixed-length representations.

**The second network** is the pattern-conditioned diffusion network for simulating a diffusion-denoising process - perturbing the pattern representations gradually by adding noise over $N$ steps (diffusion) and removing the noise to gradually recover the original representation (denoising). The diffusion process is achieved by a pre-specified procedure of incrementally adding Gaussian noise step by step, while the denoising process is approximated by a neural network that learns the removing noise at each step, i.e., the denoising gradient. Approximating the stepwise denoising gradients is equivalent to learning the mapping from a latent Gaussian space to the pattern space. Consequently, given a Gaussian noise, we can generate a pattern representation. The continuous nature of the Gaussian space implies that we can sample an infinite amount of Gaussian noise and produce corresponding new pattern representations. We build our diffusion network based on the denoising diffusion probabilistic model (DDPM) (Ho et al., 2020)[4]. In detail, we apply the following diffusion process at each step $i$ to corrupt the representation into noise:

$$q(\boldsymbol{x}^i|\boldsymbol{x}^{i-1}) = \mathcal{N}(\boldsymbol{x}^i; \sqrt{1-\beta}(\boldsymbol{x}^{i-1} - \boldsymbol{p}), \beta\boldsymbol{I}), \tag{3}$$

where $\beta$ represents the magnitude of the segments. Thereafter, we design a conditional denoising process that recovers the target segments from a prior Gaussian noise conditioned on the reference patterns over the reversed $N$ steps:

$$p_\theta(\boldsymbol{x}^{i-1}|\boldsymbol{x}^i) = \mathcal{N}(\boldsymbol{x}^{i-1}; \mu_\theta(\boldsymbol{x}^i, i, \boldsymbol{p}), \beta\boldsymbol{I}), \tag{4}$$

where $\mu_\theta$ is proportional to $\epsilon_\theta$ representing the neural network that learns the denoising gradient at each step (see Appendix C.1). Note that the superscript $i$ denotes the step in the diffusion and denoising process.

**We jointly train** the pattern-conditioned diffusion network and the scaling AE using the standard supervised learning with the segments identified in Sec. 4.1 as training data. As depicted in Fig. 4(b), the observed segments are encoded and perturbed to noise by the encoder in the scaling AE and the diffusion process in the pattern-conditioned diffusion network. The generation process, marked with dashed arrows, reverses this by denoising and decoding the segments from noise through the denoising process in our diffusion network and the decoder in scaling AE. During this process, we must ensure that (i) the diffusion and denoising gradients are consistent at each step, and (ii) the reconstruction successfully reproduces the observed segments. Therefore, the objective contains the reconstruction loss between the observed and reconstructed segments for the scaling AE and the unweighted variant of the variational lower bound (ELBO) (Ho et al., 2020) for the pattern-conditioned diffusion network:

$$\mathcal{L}(\theta) = \mathbb{E}_{\boldsymbol{x}_m}[\|\boldsymbol{x}_m - \hat{\boldsymbol{x}}_m\|_2^2] + \mathbb{E}_{\boldsymbol{x}_m^0, i, \epsilon}[\|\epsilon^i - \epsilon_\theta(\boldsymbol{x}_m^i, i, \boldsymbol{p})\|_2^2], \tag{5}$$

where $\epsilon^i$ is the noise added in the corresponding diffusion process at step $i$.

During the generation phase, new segments can be created by exclusively applying the denoising process in the pattern-conditioned diffusion network and the decoder in the scaling AE.

## 4.3 PATTERN EVOLUTION: LEARNING THE TRANSITION BETWEEN CONSECUTIVE PATTERNS

As mentioned in Sec. 3.1, we model the transition states (encompassing patterns, lengths, and magnitudes) between consecutive generated segments using a Markov chain. Once the transition

---

[4]We present the preliminary of DDPM and our network structures in Appendix C.1 and C.2, respectively.

states are determined, we obtain an evolution series of patterns, somehow addressing the irregularity in the financial time series. This ensures that the consecutive generated segments maintain the essential temporal correlations observed in real-world financial data. To capture the Markov-chain modeled temporal dynamics across patterns, we introduce a pattern evolution network $\phi$, with the network structure in AppendixD.1, to learn the temporal evolution of the states between consecutive segments[5]. More specifically, the network learns the probability of the next pattern along with its corresponding length and magnitude, given the current state (because of the Markov property):

$$(\hat{p}_{m+1}, \hat{\alpha}_{m+1}, \hat{\beta}_{m+1}) = \phi(p_m, \alpha_m, \beta_m), \tag{6}$$

where $(\hat{p}_{m+1}, \hat{\alpha}_{m+1}, \hat{\beta}_{m+1})$ denotes the next pattern and its scales in length and magnitude.

The pattern evolution network is trained to optimize the following objective:

$$\mathcal{L}(\phi) = \mathbb{E}_{\boldsymbol{x}_m}[\ell_{CE}(p_{m+1}, \hat{p}_{m+1}) + \|\alpha_{m+1} - \hat{\alpha}_{m+1}\|_2^2 + \|\beta_{m+1} - \hat{\beta}_{m+1}\|_2^2], \tag{7}$$

where $\ell_{CE}(\cdot, \cdot)$ represents the cross-entropy.

### 4.4 Putting Everything Together: Synthesizing Entire Financial Time Series

We regard patterns as the basic building blocks of generation. Accordingly, FTS-Diffusion produces synthetic time series on a pattern-by-pattern basis.

Given an initial segment sampled from the historical data, it generates the successive segments by employing the pattern generation module and the pattern evolution module iteratively, as outlined in Algorithm 2 in Appendix D.2. At each position $m$, the pattern evolution network predicts the next pattern $p_{m+1}$, its length-scaling factor $\alpha_{m+1}$, and magnitude-scaling factor $\beta_{m+1}$. With these states, the pattern generation module generates the next segment $\boldsymbol{x}_{m+1}$. The synthetic time series then grows as more segments are generated and appended. This procedure is repeated until the entire time series reaches the desired total length.

## 5 Numerical Experiments

We conduct numerical experiments to evaluate the performance of our FTS-Diffusion compared with alternatives, i.e., whether the generated data resemble real data and are useful for downstream tasks.

### 5.1 Data and Experimental Setting

We run experiments on three different types of financial assets with varying characteristics: the Standard and Poor's 500 index (S&P 500), the stock price of Google (GOOG), and the corn futures traded on the Chicago Board of Trade (ZC=F). Detailed data settings are given in Appendix E.1. In finance, it is known that the raw asset prices follow a non-stationary random walk and are not well-behaved for statistical models. Instead, the returns, i.e., closing price changes in consecutive time intervals, remain relatively constant statistical properties (such as mean and variance) over time. Thus, we compare the return series generated by FTS-Diffusion to those by representative baselines: RCGAN (Esteban et al., 2017), TimeGAN (Yoon et al., 2019), and CSDI (Tashiro et al., 2021), whose details are in Appendix E.2.

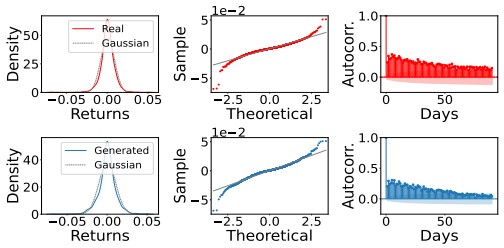

Figure 5: Stylized facts of real and generated S&P 500 over 10 years: the heavy-tailed distribution (fat tails compared to the Gaussian in density and QQ-plot, the first two columns) and decaying auto-correlations in absolute return (the last column).

---

[5]We do not estimate transitions using traditional Markov models. Our NN-based approach avoids unwieldy transition matrices, generalizes well to unseen scenarios, and handles non-linear dependencies adeptly.

Table 1: Generated return distributions compared to observed data. The KS and AD statistics are floored/capped at 0/1 and 0.01/0.25, respectively. A higher value indicates better goodness of fit. Variation in the test statistic across multiple runs is shown with a +/- range.

| Model | S&P500 | | GOOG | | ZC=F | |
|---|---|---|---|---|---|---|
| | KS | AD | KS | AD | KS | AD |
| RCGAN | $.189_{\pm.006}$ | $.073_{\pm.004}$ | $.185_{\pm.006}$ | $.068_{\pm.004}$ | $.179_{\pm.006}$ | $.065_{\pm.005}$ |
| TimeGAN | $.293_{\pm.004}$ | $.115_{\pm.006}$ | $.288_{\pm.007}$ | $.108_{\pm.005}$ | $.287_{\pm.007}$ | $.103_{\pm.005}$ |
| CSDI (Generative) | $.168_{\pm.003}$ | $.069_{\pm.002}$ | $.156_{\pm.004}$ | $.067_{\pm.003}$ | $.157_{\pm.003}$ | $.065_{\pm.003}$ |
| FTS-Diffusion | $\mathbf{.327}_{\pm.003}$ | $\mathbf{.128}_{\pm.003}$ | $\mathbf{.324}_{\pm.004}$ | $\mathbf{.119}_{\pm.002}$ | $\mathbf{.325}_{\pm.003}$ | $\mathbf{.121}_{\pm.003}$ |

## 5.2 PROPERTIES OF THE SYNTHETIC TIME SERIES

The synthetic financial time series should inherit the stylized facts (Cont, 2001; Barberis & Shleifer, 2003) of asset returns, and resemble the distribution of observed data to a high degree of fidelity.

**Stylized facts of financial time series.** The empirical properties of financial time series have been studied extensively in the literature, which is often referred to as stylized facts (Cont, 2001; Barberis & Shleifer, 2003). The empirical studies reveal that asset returns have heavy tails, and the autocorrelation of absolute returns decays slowly over time. We assess whether the synthetic time series adhere to these stylized facts in Fig. 5. Indeed, the synthetic series exhibit significant heavy tails in their distribution and gradual decay in the autocorrelation of absolute returns, conforming to the aforementioned stylized facts. These results suggest that our approach is capable of generating synthetic financial time series that preserve the essential properties of observed data.

**Distribution comparison.** We also evaluate the discrepancy between the distribution of the synthetic time series and that of observed data, using the Kolmogorov–Smirnov (KS) test and the Anderson–Darling (AD) test as evaluation metrics. These tests estimate the goodness of fit between the synthesized distribution and the distribution of actual returns. For both tests, a larger test statistic indicates a higher degree of similarity between the distributions. The KS test is more sensitive to differences in the center of the distribution, whereas the AD test is more aware of the tails of the distribution. Table 1 demonstrates that our FTS-Diffusion learns a quantitatively closer distribution to the observed data, compared to other baselines. This result further confirms the efficacy of our approach in generating financial time series that resemble the observed data. More quantitative results using other metrics are included in Appendix E.3.

## 5.3 DOWNSTREAM PREDICTION ANALYSIS OF THE SYNTHETIC TIME SERIES

We expand "Training on Synthetic, Test on Real" (Esteban et al., 2017; Jordon et al., 2018) and design two new settings to evaluate the usefulness of the synthetic data for downstream tasks. Specifically, we focus on the task of prediction and implement an LSTM-based downstream predictive model. This structure is a prevalent choice in the literature (Yoon et al., 2019; Jeon et al., 2022; Remlinger et al., 2022). The downstream model is employed to predict the next data point in the series, using the 64 previous historical values as input (see Appendix E.4 for the additional five-day ahead prediction). We compute the mean absolute percentage error (MAPE) averaged over multiple runs.

**Training on Mixture, Test on Real (TMTR).** In this setting, we train the downstream predictive model on a dataset that combines observed and synthetic data in different proportions. For instance, a dataset with a mixing proportion of (30%, 70%) would be composed of 30% of data sampled from the observed data and 70% of data synthesized by the generative model. We test the predictive model on the test set sampled from the observed data which had not been seen by the generative model. If the synthetic data resemble the observed data, the predictive power of the downstream model trained on datasets with different mixing proportions should remain similar. Fig. 6(a) shows the results of the TMTR experiment for the one-day forecast on the three assets. The predictive accuracy is remarkably consistent across all mixing proportions, when synthetic data are generated using FTS-Diffusion. In comparison, the predictive accuracy deteriorates (large MAPEs) as the proportion of observed data decreases, when synthetic data are generated using RCGAN, TimeGAN, or CSDI.

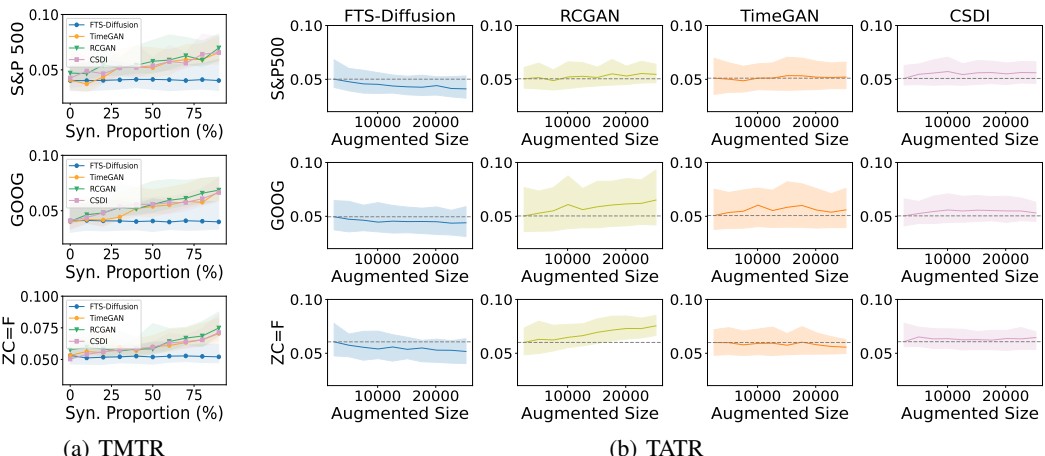

(a) TMTR                    (b) TATR

Figure 6: Prediction errors of the downstream model trained under the TMTR and TATR settings. Our FTS-diffusion maintains a comparable level of prediction accuracy across all mixing proportions of synthetic data and reduces the prediction errors by augmenting the observed dataset. Solid lines and shaded bands in each subfigure represent the average error and the 95% confidence interval over multiple runs, respectively. Dashed lines in each TATR test mark the initial prediction errors.

Thus, FTS-Diffusion is capable of generating synthetic time series sufficiently similar to actual data to uphold the performance of a downstream prediction task, whereas other models cannot.

**Training on Augmentation, Test on Real (TATR).** We initialize the training set with limited observed data. We then iteratively append additional synthetic data and evaluate the resulting performance of the downstream predictive model for a one-day ahead forecast. The results in Fig. 6(b) show a clear downward trend in the prediction error as more synthetic data from FTS-Diffusion is added to the training set. Appending 100 years of synthetic data reduces the MAPE by 17.9%, 15.3%, and 17.4% on the three assets, respectively. In contrast, the prediction error either increases or largely remains the same when synthetic data are generated by other baselines. These results indicate that FTS-Diffusion can effectively alleviate the problem of data shortage by augmenting the training set with sufficient synthetic samples. Supplementary experiments are provided in Appendix E.

## 6 CONCLUDING REMARK

We present *FTS-Diffusion*, a generative framework, for synthesizing financial time series with irregular and scale-invariant patterns. We break down the challenging financial time series generation into a *pattern recognition-generation-evolution* scheme. To facilitate this process, we design three dedicated modules: (i) a pattern recognition module leveraging our proposed SISC algorithm carefully designed to identify these patterns, (ii) a pattern generation module using a diffusion-based network to synthesize the segments of patterns, and (iii) a pattern evolution network to assemble generated segments with proper temporal evolution between consecutive patterns. Experimental results confirm the effectiveness of FTS-Diffusion in synthesizing financial time series that resemble observed data in distribution and their usefulness for downstream tasks. To the best of our knowledge, this is the first work in generating intricate yet crucial time series that encompass irregular and scale-invariant patterns, holding the potential for diverse applications across domains beyond finance.

This work offers a new perspective on complex (financial) time series generation from its irregular and scale-invariant properties. A promising direction for future research would be to extend our work to more challenging problem settings, e.g., the multivariate modeling that encompasses interactive dependencies across multiple time series. Our approach can handle potential distribution shifts arising from changes in the number of patterns, and an extension may be able to address shifts in transitions between patterns. One could also strengthen the theoretical and empirical guarantee of the generation quality. We leave these ideas for future research.

ACKNOWLEDGEMENT

The authors wish to dedicate this work to the memory of Prof. Duan Li, whose insights and guidance have been instrumental to the early stage of this work.

This work is supported in part by General Research Funds from Research Grants Council, Hong Kong (Project No. 11200223, 21500422, and 11500823), an InnoHK initiative, The Government of the HKSAR, Laboratory for AI-Powered Financial Technologies, and a Shenzhen-Hong Kong-Macau Science & Technology Project (Category C, Project No. SGDX20220530111203026). The authors would also like to thank Mark Huang, Blair Hull, Ray Iwanowski, Alexander James, Carrie Wang, Qi Wu, Hao Pan, and the anonymous reviewers for their helpful comments.

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

## A    FURTHER COMPARISON OF FINANCIAL TIME SERIES AND OTHER REGULAR SERIES

As discussed in Sec. 1, financial time series exhibit a more complex distribution than other time series. In particular, the inherent patterns in financial time series are irregular and scale-invariant. We have thoroughly discussed the scale-invariance in Sec. 1 and 4. To better illustrate the irregular property of financial time series, we display the distributions of financial time series and other time series sampled at different frequencies by ridge plots as Figure 7. Financial time series clearly exhibit distinct distributions for each month in 2022, as well as for each year spanning from 2017 to 2021. In contrast, other time series show similar distributions regardless of different sampling frequencies. The results exemplify that the distributions of the subsequences in financial time series tend to be diverse over time, reflecting the irregular property.

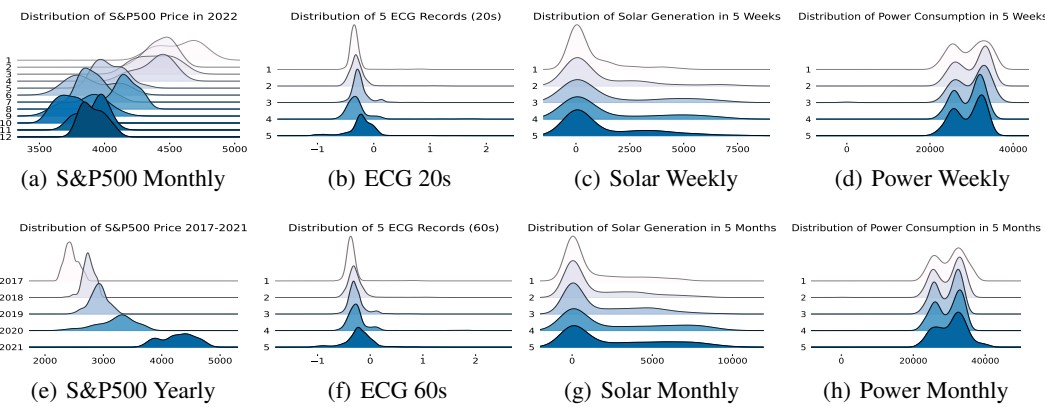

Figure 7: Distributions of time series sampled with different frequencies: (a,e) S&P500 (finance), (b,f) ECG (medical), (c,g) solar generation (renewable), and (d,h) power consumption (smart grid). Financial time series, such as S&P 500, exhibit different distributions sampled at different frequencies. Whereas, the distributions of other time series are similar regardless of the sampling frequency, respectively.

## B    MORE INFORMATION ABOUT OUR PATTERN RECOGNITION MODULE

### B.1    SUPPLEMENTARY TECHNICAL DETAILS OF OUR SISC ALGORITHM

The pseudo-code of our SISC algorithm is presented as Algorithm 1. As introduced in Sec. 4.1, SISC is performed with two main stages, (i) initializing the cluster centroids (Cluster Initialization in Algorithm 1) and (ii) segmenting and clustering the subsequences into $K$ clusters using a greedy strategy (Greedy Segmentation and Clustering in Algorithm 1).

In numerical experiments in Sec. 5, leveraging domain knowledge in finance, we set the minimum and maximum segment lengths as 10 and 21, respectively, focusing on the atom-like short-term patterns commonly observed (Lo et al., 2000). Applying the elbow method (Thorndike, 1953), we empirically determine the $K$'s for three financial assets, which are 14, 11, and 11, respectively.

### B.2    VERIFICATION OF LEARNED PATTERNS WITH PREDEFINED COUNTERPARTS IN FINANCE

Employing the pattern recognition module, we successfully learn patterns in financial time series in a data-driven manner. The majority of the learned patterns are consistent with predefined ones with particular shapes and formulaic definitions from technical analysis (Lo et al., 2000). Table 2 presents the consistent patterns along with their terminologies from technical analysis. The learned pattern, as shown in the first row and second column of Table 2, completely coincides with the classical Inverse Head-and-Shoulders (IHS) via visual observation. Despite some sight noise, the other learned patterns exhibit similar shapes to the predefined ones. Surprisingly, the learned patterns are sensitive

---

**Algorithm 1** SISC Algorithm

---

**Require:** Time series $\boldsymbol{X}$, pre-determined number of clusters $K$, minimum and maximum subsequence length $l_{\min}, l_{\max}$, maximum iterations $max\_iters$
  1: $\mathcal{P} \leftarrow \emptyset$
  2: Prepare candidate centroids $\{\boldsymbol{X}_{t:t+l_{\max}}\}_{t=0}^{T-l_{\max}}$
  3: Randomly select the first centroid $p_0$ from the candidates
  4: $\mathcal{P}.append(p_0)$
  5: **while** $\mathcal{P}.size < K$ **do**                    ▷ Cluster Initialization
  6:     Compute the distance to the nearest chosen centroid for each remaining candidate
  7:     Set the probability of each candidate proportional to the above distance
  8:     Randomly select the next centroid $p_k$ with the above probability
  9:     $\mathcal{P}.append(p_k)$
 10: **end while**
 11: $iter \leftarrow 0$
 12: **while** $iter < max\_iters$ **do**                  ▷ Greedy Segmentation and clustering
 13:     $\mathcal{S} \leftarrow \emptyset$
 14:     $t \leftarrow 0$
 15:     **while** $t < T$ **do**
 16:         $l^* \leftarrow \arg\min_{l \in [l_{\min}, l_{\max}]} DTW(X_{t:t+l}, \boldsymbol{p}), \forall \boldsymbol{p} \in \mathcal{P}$
 17:         $\mathcal{S}.append(l^*)$
 18:         $t \leftarrow t + l^*$
 19:     **end while**
 20:     $iter \leftarrow iter + 1$
 21:     Update $\mathcal{P}$
 22: **end while**
 23: Return $\mathcal{P}, \mathcal{S}$

---

to minor changes in temporal dynamics. For instance, the Descending Rectangle (DR), the last one in the second column, is slightly different in its trend. Our pattern recognition module is capable of distinguishing such slight differences. This cross-verification indicates that the unsupervised algorithm is able to identify genuine recurring patterns and significantly raises the level of confidence in our data-driven approach.

Moreover, we have revealed some intriguing findings. Our pattern recognition module focuses on identifying the irregular and scale-invariant patterns in the financial time series, while technical analysis defines the patterns as trading signals to forecast future trends. Therefore, if the future trend predicted by technical analysis is exactly highly correlated with the predefined patterns, it is possible for them to emerge together in our clustering. In other words, our learned patterns may incorporate the future trend into the corresponding predefined patterns from technical analysis. As an example, the Rectangle Top (RTOP), the last one in the first column indicates a decreasing future trend in technical analysis, and our result covers such a declining curve. This provides new insights into technical analysis, by revealing patterns that combine the trading signal with future trends.

However, not all learned patterns match predefined patterns. We are faced with the dilemma of whether to include these unmatched patterns in data generation. As we will show later, downstream experiments reveal the benefit of including unmatched patterns. While there is a long tradition of technical analysis, the predefined patterns rely on human identification. The unmatched patterns are likely capturing important components of financial data, yet too nuanced for humans to see.

## B.3 INVESTIGATION WITH SIMULATED TIME SERIES DATA

In this investigation, we explore the capabilities of our pattern recognition module in detail, in particular for the proposed SISC algorithm. To effectively demonstrate its functionality, we conduct a series of experiments using various simulated time series data that we have manually created. These simulated data comprise multiple scale-invariant patterns that we define, exhibiting varying lengths and magnitudes. The key advantage of these simulated data is that we have full control over the data-generating process. This control allows us to know the ground truth against which we can

Table 2: Patterns learned by FTS-Diffusion are consistent with predefined ones from technical analysis (Lo et al., 2000).

| Terminology | Standard | Learned | Terminology | Standard | Learned |
|---|---|---|---|---|---|
| Head&Shoulders (HS) | | | Inverse Head&Shoulders (IHS) | | |
| Triangle Tops (TTOP) | | | Triangle Bottoms (TBOT) | | |
| Double Tops (DTOP) | | | Double Bottoms (DBOT) | | |
| Ascending Rectangle (AR) | | | Descending Rectangle (DR) | | |
| Rectangle Top (RTOP) | | | | | |

compare and evaluate the model performance. Therefore, this setup enables us to rigorously evaluate the module under a variety of controlled scenarios.

More specifically, we construct three distinct simulated time series, each being designed with different configurations of recurring scale-invariant patterns. These configurations are as follows: (i) one pattern only, and (ii) multiple patterns. For each configuration, we establish the corresponding standard pattern(s). Subsequently, each pattern is transformed into diverse segments that maintain similar shapes but exhibit varying lengths and magnitudes. This transformation involves stretching the pattern to variable lengths and magnitudes as well as integrating a certain degree of noise. These segments are then combined together to form an entire time series with a length of 10000 data points, following a pre-specified transition between consecutive patterns. Consequently, we keep hold of full control over the simulated time series, including the ground-truth patterns, the number of patterns, the segmentations, and the transition states.

We assess the performances of our SISC algorithm on these simulated data from two primary perspectives, whether it accurately learns the cluster centroids that represent the identified patterns and whether it learns the correct segmentation relative to the ground truth. These also constitute the key objectives of our pattern recognition module. The ground truth centroid of each cluster is the barycenter of segments transformed from the same corresponding pattern. To quantitatively measure the discrepancy between each centroid $c_{sisc}$ learned by SISC and the ground truth $c_{real}$, we design a per-unit DTW error as $DTW(c_{real}, c_{sisc})/L$ over a normalized space, where $L$ denotes the length of ground truth centroids. In this context, the normalized space implies that the magnitude of each point in the centroid is rescaled to fall within the range of [0,1]. This metric describes the average deviation per unit length between the learned centroid and the ground truth, as measured in the time-warped space. A lower value suggests a higher degree of similarity between the learned centroid and the ground truth. The upper bound of this metric is 1, which represents complete dissimilarity between the learned and true centroids. For evaluating the segmentation, we utilize the Jaccard similarity coefficient, also known as the Intersection over Union (IoU), to compare the segmentation $s_{sisc}$ learned by SISC and the ground truth $s_{real}$. It is defined as, $|s_{real} \cap s_{sisc}|/|s_{real} \cup s_{sisc}|$, the size of the intersection divided by the size of the union of the sample sets, which is floored and capped at 0 and 1, respectively. A higher statistic means a greater similarity between the learned segmentation and the ground truth, indicating a more accurate segmentation by the SISC algorithm. Next, we will present the experimental results for each simulated time series.

*One pattern only.* This setup is designed to verify the ability of our SISC algorithm to correctly identify and segment a single pattern from the simulated time series. In this scenario, the simulated data has only one simple pattern, depicted as the standard pattern in Fig.8(b). This standard pattern is manipulated into devise segments, each with distinct lengths and magnitudes following pre-specified transitions between consecutive segments. These segments are then merged to construct a complete time series, as illustrated in Fig.8(a). After performing SISC on this simulated time series, the ground truth centroid of these segments (Fig.8(c)) and the counterpart learned by SISC (Fig.8(d)) show a significant degree of similarity, achieving a very low per-unit DTW error of 0.009. Furthermore, the Jaccard similarity coefficient of the resulting segmentation is 0.938, suggesting that 93.8% of the points in the simulated time series are correctly segmented. These results verify that our SISC algorithm successfully identifies the target pattern and correctly segments the simulated time series containing one pattern only.

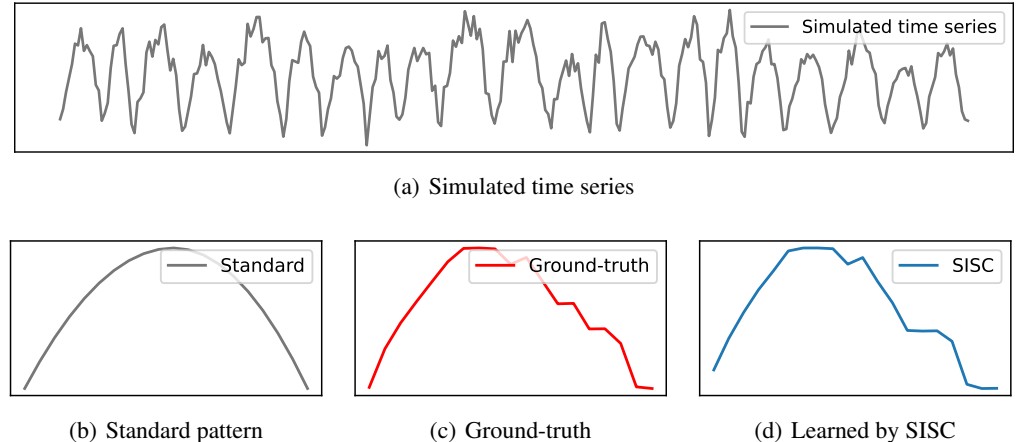

(a) Simulated time series

(b) Standard pattern      (c) Ground-truth      (d) Learned by SISC

Figure 8: Investigation of SISC: one-pattern scenario. (a) Simulated time series containing one scale-invariant pattern only; (b) predefined standard pattern; (c) ground-truth centroid; (d) centroid learned by SISC.

*Multiple patterns.* Our focus on this configuration is to further validate the effectiveness of our SISC algorithm in learning the simulated time series that involves four patterns in Fig. 9. The data-generating process is the same as that in the one-pattern scenario but produces a variety of segments corresponding to multiple patterns. The learned centroids by SISC are comparable to the ground truth, with an average per-unit DTW error of 0.01 over four patterns. Meanwhile, the learned segmentation achieves a Jaccard similarity coefficient of 0.784. This score indicates that a significant majority of data points are well placed into the correct intervals. The points that are incorrectly segmented are predominantly located on the boundaries of the segments. Importantly, these boundary misclassifications do not significantly impact the general shape of the corresponding segments. Therefore, despite these minor discrepancies at the segment boundaries, our SISC algorithm demonstrates its robustness when applied to simulated data consisting of multiple patterns.

We believe this investigation offers valuable insights into the pattern recognition module, particularly our proposed SISC algorithm, and confirms its potential applicability and reliability in more complex real-world scenarios.

## C  MORE INFORMATION ABOUT OUR PATTERN GENERATION MODULE

### C.1  PRELIMINARY OF DENOISING DIFFUSION PROBABILISTIC MODEL

Diffusion models incorporate a forward diffusion process $q(\boldsymbol{x}^i|\boldsymbol{x}^{i-1})$ that gradually corrupts the target data by adding noise over $N$ steps and a backward denoising process $p_\theta(\boldsymbol{x}^{i-1}|\boldsymbol{x}^i)$ that learns the reverse procedure to recover the target data, where $i \sim \mathcal{U}(1, 2, ..., N)$ denotes the $i$-th diffusion step. The diffusion process is typically pre-specified:

$$q(\boldsymbol{x}^{1:N}|\boldsymbol{x}^0) := \prod_{i=1}^{N} q(\boldsymbol{x}^i|\boldsymbol{x}^{i-1}), \tag{8}$$

$$q(\boldsymbol{x}^i|\boldsymbol{x}^{i-1}) := \mathcal{N}(\boldsymbol{x}^i; \sqrt{1 - \sigma^i}\boldsymbol{x}^{i-1}, \sigma^i \boldsymbol{I}). \tag{9}$$

The denoising process is approximated with a neural network as follows:

$$p_\theta(\boldsymbol{x}^{0:N}) := p(\boldsymbol{x}^N) \prod_{i=1}^{N} p_\theta(\boldsymbol{x}^{i-1}|\boldsymbol{x}^i), \tag{10}$$

$$p_\theta(\boldsymbol{x}^{i-1}|\boldsymbol{x}^i) := \mathcal{N}(\boldsymbol{x}^{i-1}|\mu_\theta(\boldsymbol{x}^i, i), \sigma^i \boldsymbol{I}), \tag{11}$$

where the neural network $\theta$ learns the noise gradient between steps and $\sigma^i$ is a predefined parameter.

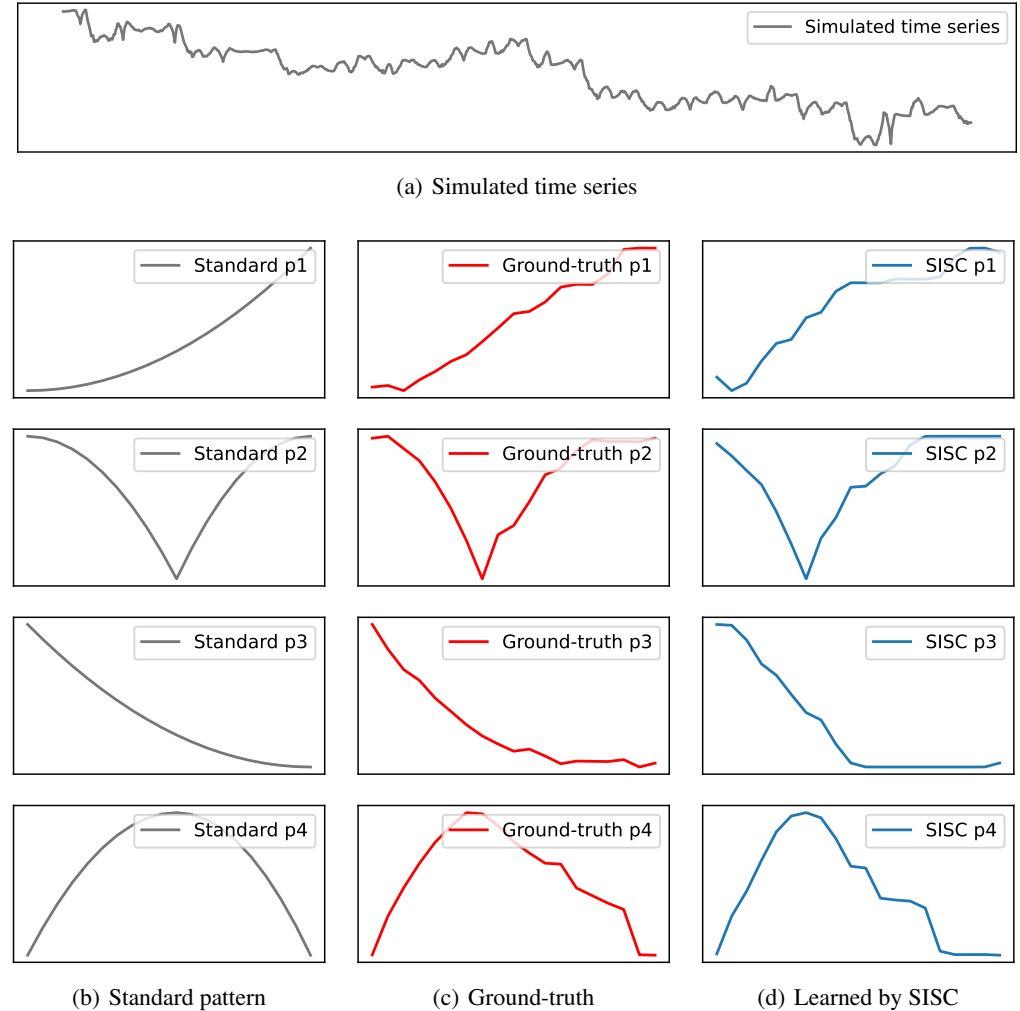

Figure 9: Investigation of SISC: multiple-patterns scenario. (a) Simulated time series containing multiple scale-invariant patterns; (b) predefined standard patterns; (c) ground-truth centroids; (d) centroids learned by SISC.

The denoising diffusion probabilistic model (DDPM) (Ho et al., 2020) introduces a nice property that enables us to sample $\boldsymbol{x}^i$ at an arbitrary step $i$ during the diffusion process, using the reparameterization trick. Let $\upsilon^i = 1 - \sigma^i$ and $\bar{\upsilon}^i = \prod_{j=1}^{i} \upsilon^i$, the resulting closed form is:

$$q(\boldsymbol{x}^i|\boldsymbol{x}^0) = \mathcal{N}(\boldsymbol{x}^i; \sqrt{\bar{\upsilon}^i}\boldsymbol{x}^0, (1 - \bar{\upsilon}^i)\boldsymbol{I}). \tag{12}$$

Using the same property, we can also obtain:

$$\mu_\theta(\boldsymbol{x}^i, i) = \frac{1}{\sqrt{\upsilon^i}}(\boldsymbol{x}^i - \frac{1 - \upsilon^i}{\sqrt{1 - \bar{\upsilon}^i}}\epsilon_\theta(\boldsymbol{x}^i, i)). \tag{13}$$

Therefore, (Ho et al., 2020) demonstrated that training diffusion models with the following simplified objective empirically yields better results:

$$\mathcal{L} = \mathbb{E}_{i \sim [1,N], \boldsymbol{x}^0, \epsilon^i}[\|\epsilon^i - \epsilon_\theta(\boldsymbol{x}^i, i)\|^2]. \tag{14}$$

## C.2 IMPLEMENTATION DETAILS OF OUR PATTERN-CONDITIONED DIFFUSION NETWORK AND SCALING AUTOENCODER

Our pattern-conditioned diffusion network utilizes six residual temporal convolutional (TCN) blocks to capture the internal temporal dynamics within pattern segments. Each block mainly comprises two temporal convolution layers. Time embeddings for each diffusion step are constructed with a fully-connected layer positioned at the top of each block. We set the number of diffusion steps to $N = 100$.

Our scaling AE can be implemented with two layers of LSTMs or GRUs.

We jointly train these two networks following the procedure in Sec. 4.2 using the Adam optimizer with a learning rate of $5e - 04$. We set the batch size to 32. The hyper-parameters are determined empirically following the common techniques in the literature.

# D MORE INFORMATION ABOUT OUR PATTERN EVOLUTION MODULE

## D.1 IMPLEMENTATION DETAILS OF THE PATTERN EVOLUTION NETWORK

In practice, we treat the modeling of the next pattern $p$ as a multi-category classification, while the learning of the length-scaling factor $\alpha$ and the magnitude-scaling factor $\beta$ as regression. Note that treating the estimation of the length as a classification task is also feasible. Hence, our pattern evolution network models the Markov transition of state $(p, \alpha, \beta)$ between consecutive segments by a fully-connected neural network with three corresponding outputs. We train this network using the Adam optimizer with a learning rate of $4e - 04$ over 1000 epochs. The hyper-parameters are determined empirically.

## D.2 PSEUDO-CODE OF THE SAMPLING PROCESS

We provide the pseudo-code of the sampling process in our FTS-Diffusion as Algorithm 2. We commence the creation of a new synthetic time series by initializing the first segment, which is sampled from the observed data. After the initialization, subsequent segments are produced iteratively through the following procedure. In each iteration, the transition states of the next segment are first predicted using the pattern evolution module $\phi$. With these states, the next segment is generated by the pattern generation module $\theta$. This newly generated segment is then appended to the synthetic time series. This iterative process is repeated until the synthetic time series reaches the desired length.

---

**Algorithm 2** Data synthesizing procedure incorporating the pattern generation module and pattern evolution module.

**Require:** Pattern generation module $\theta$, pattern evolution module $\phi$, latent patterns $\mathcal{P}$, terminal series length $T$

1: $\hat{X} \leftarrow \emptyset$
2: Initialize $x_0 \in X$
3: $\hat{X}.append(x_0)$
4: $m \leftarrow 0$
5: **while** $len(\hat{X}) < T$ **do**
6: $\quad p_m, \alpha_m, \beta_m \leftarrow TransitionStates(x_m)$
7: $\quad (p_{m+1}, \alpha_{m+1}, \beta_{m+1}) \leftarrow \phi(p_m, \alpha_m, \beta_m)$
8: $\quad x_{m+1} \leftarrow \theta(p_{m+1}, \alpha_{m+1}, \beta_{m+1})$
9: $\quad \hat{X}.append(x_{m+1})$
10: $\quad m \leftarrow m + 1$
11: **end while**
12: Return $\hat{X}$

---

# E  SUPPLEMENTARY DETAILS ON EXPERIMENTS

## E.1  DATA SETTINGS

As introduced in the main paper, we conduct our experiments on three assets: the Standard and Poor's 500 index (S&P 500), the stock price of Google (GOOG), and the corn futures traded on the Chicago Board of Trade (ZC=F). The S&P 500 data covers the period from 1980-01-01 to 2020-01-01. The data for GOOG spans from 2005-01-01 to 2020-01-01. And the ZC=F data ranges from 2001-01-01 to 2020-01-01.

We employ an 80/20 train-test split strategy, using the first 80% for training and the remaining 20% for testing. Importantly, neither FTS-Diffusion nor downstream models in our subsequential experiments have seen the test sets during the training phase, and all of our evaluation is on an out-of-sample basis.

## E.2  IMPLEMENTATION DETAILS OF BASELINES

**RCGAN.** In our experiment, we apply the conditional version of RCGAN (Esteban et al., 2017), with time information (e.g., date) as the input conditions.

**TimeGAN.** TimeGAN (Yoon et al., 2019) first learns an embedding of the target data with an autoencoder network and then models the latent distribution with a GANs network. Consequently, the synthetic time series can be constructed by using the decoder to expand the generator output into the original data space. In the original paper, TimeGAN takes multi-dimensional prices (including the volume and high, low, opening, closing, and adjusted closing prices) as inputs. However, our experimental settings focus on univariate time series. For a fair comparison, we implement the TimeGAN following the original paper but with a one-dimensional input of the return rate of daily closing prices.

**CSDI.** CSDI (Tashiro et al., 2021) is proposed for time series imputation. As indicated in its original paper, the unconditional variant of CSDI can be utilized for time series generation. Thus, we implement the unconditional variant of CSDI for financial time series generation.

All of the baselines are designed to synthesize the identical-interval samples. Thus, we partition the entire financial time series into equal-length subsequences using the classical segmentation approach. The length of each subsequence is set as 21, aligning with the setting of maximum length in our FTS-Diffusion.

## E.3  QUANTITATIVE ANALYSIS USING ADDITIONAL METRICS

In addition to the Kolmogorov–Smirnov (KS) test and the Anderson–Darling (AD) test employed in our primary experiments, there exist numerous other effective tests to evaluate the 'goodness of fit' between the distributions of different time series. Here, we include two more widely-used metrics: the nonparametric Epps-Singleton (ES) test and the Wasserstein Distance (W). The ES test, frequently used in the field of econometrics, is sometimes preferable to the KS test in instances that do not assume a continuous distribution. A larger ES statistic represents a higher similarity between the distributions of the two samples. The Wasserstein distance measures the minimum amount of probability mass that needs to be moved between two probability distributions. A smaller distance indicates a closer match between the distributions. Table 3 presents a comparison of generated returns using these metrics. According to these distribution tests, our FTS-Diffusion outperforms other baselines as well.

Moreover, we incorporate several fundamental metrics prevailing in finance, including the Sharpe ratio (SR), Sortino ratio (SoR), and Calmar ratio (CR). The Sharpe ratio measures the average return earned in excess of the risk-free rate per unit of volatility, which helps investors understand the return of an asset compared to its risk. The Sortino ratio, akin to the Sharpe ratio but with a focus on downside risk, evaluates the excess return against its downside deviation. The Calmar Ratio compares the average compounded rate of return to the maximum drawdown risk, offering insight into the potential losses an asset might experience. These summary statistics represent fundamental characteristics of financial asset returns. If the synthetic time series closely resemble the observed data, these ratios of the synthetic data would align closely with those of the real data. In other

Table 3: Generated return distributions compared to observed data using more metrics. Variation in the test statistic across multiple runs is shown with a +/- range.

| Model | S&P500 | | GOOG | | ZC=F | |
|---|---|---|---|---|---|---|
| | ES ↑ | W ↓ | ES ↑ | W ↓ | ES ↑ | W ↓ |
| RCGAN | .004 ±.0007 | .009 ±.0005 | .005 ±.0006 | .009 ±.0005 | .004 ±.0006 | .009 ±.0005 |
| TimeGAN | .006 ±.0005 | .005 ±.0004 | .005 ±.0006 | .005 ±.0004 | .005 ±.0005 | .005 ±.0005 |
| CSDI | .004 ±.0003 | .005 ±.0003 | .004 ±.0004 | .006 ±.0003 | .005 ±.0003 | .006 ±.0003 |
| FTS-Diffusion | **.008** ±.0003 | **.003** ±.0005 | **.007** ±.0003 | **.003** ±.0005 | **.008** ±.0003 | **.003** ±.0004 |

words, the discrepancies in these statistics between the actual returns of the financial asset and their comparable generated counterparts should be relatively minor. We denote these discrepancies between the actual assets and synthetic data as $\Delta_{SR}$, $\Delta_{SoR}$, and $\Delta_{CR}$, respectively. For instance, $\Delta_{SR}$ is calculated as the absolute difference between the Sharpe ratio of the actual data and that of the synthetic data, i.e., $\Delta_{SR} = |SharpeRatio(actual) - SharpeRatio(synthetic)|$. Table 4 presents the results of differences in these metrics between the actual data and the generated synthetic data. The synthetic data generated by our FTS-Diffusion yield the lowest deviation, thereby exhibiting the most comparable statistics of the fundamental metrics closely aligned with those of the actual data. This provides further evidence to support that our FTS-Diffusion replicates the fundamental characteristics of the actual financial assets.

Table 4: Difference in Sharpe ratio, Sortino ratio, and Calmar ratio between the generated returns and the actual returns of the corresponding financial assets.

| Model | S&P500 | | | GOOG | | | ZC=F | | |
|---|---|---|---|---|---|---|---|---|---|
| | $\Delta_{SR}$ | $\Delta_{SoR}$ | $\Delta_{CR}$ | $\Delta_{SR}$ | $\Delta_{SoR}$ | $\Delta_{CR}$ | $\Delta_{SR}$ | $\Delta_{SoR}$ | $\Delta_{CR}$ |
| RCGAN | 0.141 | 0.214 | 15.659 | 0.149 | 0.252 | 22.784 | 0.178 | 0.316 | 30.372 |
| TimeGAN | 0.111 | 0.189 | 12.775 | 0.177 | 0.249 | 22.067 | 0.136 | 0.203 | 26.450 |
| CSDI | 0.133 | 0.186 | 17.424 | 0.178 | 0.273 | 24.611 | 0.152 | 0.241 | 31.188 |
| FTS-Diffusion | **0.019** | **0.034** | **0.294** | **0.020** | **0.037** | **0.302** | **0.069** | **0.092** | **0.614** |

### E.4 SUPPLEMENTARY DOWNSTREAM EXPERIMENTS

For comparative purposes, we utilized a naive prediction approach as a benchmark. This method simply uses the values from the previous day to make its predictions. In this context, the mean absolute percentage errors (MAPEs) for the three datasets are 0.046, 0.053, and 0.057, respectively. These results serve as a baseline for evaluating the performance of more sophisticated forecasting models.

In Sec. 5.3, we have demonstrated the effectiveness of the synthetic time series generated by our FTS-Diffusion in the downstream experiments focused on one-day ahead prediction. To further substantiate our approach, we extend our evaluation to a more complex task of multiple-day ahead prediction. For this extended evaluation, we employ the same LSTM-based neural networks as the downstream prediction model. Fig. 10 shows the results of the TMTR and TATR experiments for five-day forecasts of S&P 500. Our FTS-Diffusion successfully maintains its performance, proving its robustness and versatility.

### E.5 DIFFERENT SETTINGS OF TRAIN/TEST SPLIT AND ROLLING WINDOW

Throughout the main body of our paper, we have evaluated our FTS-Diffusion using an 80/20 train/test split strategy. To ensure the robustness of our model and to further investigate its performance under different circumstances, we extend our evaluation to include 70/30 and 60/40 train/test split strategies. In addition to the train/test split strategies, we further probe the robustness of our model by utilizing rolling windows of different sizes for testing, particularly sizes 32 and 48. This allows us to verify that the downstream model maintains stable performance when trained on different temporal slices of synthetic data.

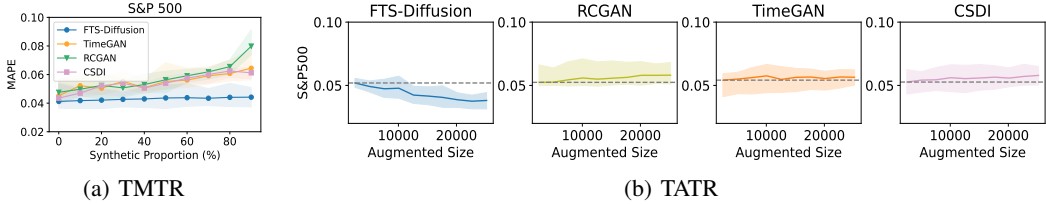

(a) TMTR             (b) TATR

Figure 10: Prediction errors of the downstream model for five-day ahead prediction trained under the TMTR and TATR settings. Our FTS-Diffusion successfully maintains its performance as in the one-day head prediction.

The outcomes of these extended downstream experiments, which utilize additional train/test split strategies and diverse rolling window sizes, are presented in Fig. 11. As more synthetic data generated by our FTS-Diffusion method are incorporated into the dataset, prediction errors are observed to decrease. This result demonstrates that augmenting the training dataset with synthetic data produced by our FTS-Diffusion method indeed bolsters the training process and the subsequent performance of downstream models. Such outcomes underscore the versatility and robustness of our FTS-Diffusion approach across various operational contexts.

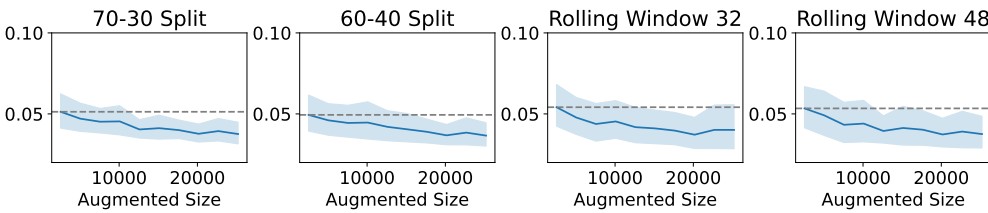

Figure 11: Prediction errors of the downstream model trained on the augmented dataset with synthetic data generated by our FTS-Diffusion under TATR using different train/test split strategies and rolling windows.

These results demonstrate that our approach maintains its capability to generalize across different train/test splits.

### E.6  CONPLEXITY AND RUNTIME ANALYSIS

We conduct an analysis comparing the computational complexity and runtimes of our approach against those of the baselines, with the results provided in Table 5. In this context, the term 'computational complexity' refers to the number of parameters in the neural networks. The 'runtimes' contain the times taken for training the model (training runtime) and generating the synthetic time series (inference runtime). Due to its multi-module nature, our FTS-Diffusion has a higher complexity. This represents a trade-off between high fidelity and complexity. Our model achieves superior generation quality at the expense of somewhat increased complexity.

Table 5: Comparison of computation complexity and runtime.

| Model | Number of Parameters | Training Runtime | Inference Runtime |
|---|---|---|---|
| RCGAN | 31190 | 23 min. | 1.72 sec. |
| TimeGAN | 65302 | 49 min. | 1.86 sec. |
| CSDI | 143284 | 97 min. | 2.61 sec. |
| FTS-Diffusion | 171209 | 131 min. | 4.27 sec. |

