# OpenReview forum: "Generative Learning for Financial Time Series with Irregular and Scale-Invariant Patterns"
_ICLR.cc/2024/Conference — ICLR 2024 spotlight_

### Official Review · Reviewer_SkxH · 2023-10-31

**Soundness:** 2 fair
**Presentation:** 3 good
**Contribution:** 2 fair
**Rating:** 8
**Confidence:** 4

**Summary:**

Learning from financial time series data is considered challenging due to limited data availability.
The present work proposes FTS-Diffusion, a generative framework to select features and model intrinsic patterns of univariate financial time series data with the aim to overcome data shortage. The model consists of the following three building blocks (1) recognition of irregular and scale-invariant patterns in each time series sample (2) generation of synthesized segments of patterns and (3) evolving of the latter over time.

**Strengths:**

This method introduces a pattern-centric approach including clustering techniques with well-known variational inference techniques and solving a classification+regression task.  It is original to the best of my knowledge. Further, the paper is well positioned to existing work, the content is well organized and, to the best of my knowledge, has no spelling or grammatical flaws. Empirical evidence is accessed through experiments on three real-world financial assets; the results are state-of-the-art. I appreciate Figure 3 since it provides a concise overview of the proposed methods.

**Weaknesses:**

1. (Pattern Conjecture) The reasons given to support the claim that univariate financial time series data (in general) embody recurring complex patterns over time appear vague and lack thorough evaluation. For example, the authors consider only a single financial data set (i.e., the S&P 500 index), and the analytical approach seems very fine-grained and not very general. It would be helpful to have insights into the clustering results, i.e., indicating the actual number of classes as well as the number of items per class, or showing multiple centers, etc. And even though Appendix A/B broadens some aspects there remains a lack of rigor in the arguments/results.

2. (Pattern Evolution) I apologize, but I lack understanding of the learning aspects of the pattern evolution network $\phi$. How do you extract the information used from the training data? I assume my question refers to the *Info()* function in $\texttt{line 6} \$ of *Algorithm 2*. Unfortunately, I can't find any more detailed information about it.

3. (Evaluation and Reproducibility)
Overall, I am not convinced that the proposed pattern-centric approach really adds complementary information to learning financial time series data. I suppose this is primarily due to deficiencies in the evaluation of the pattern recognition module. I also miss a detailed description of the included datasets and results reproducibility is very limited due no code submission.

**Questions:**

1. (p.3) I suppose the length of the entire time series is $T = \sum_{m=1}^{M} t_m - t_{m-1}$ ?
2. (Eq. (3)) $\beta \leq 1$ ?
3. (p.9 Training on Augmentation) In case of a *one-day* ahead forecast I would have expected comparison to a very *naive* approach, e.g., using the value of the previous day as prediction; especially in reference to the Markov property. Can you additionally include a multiple-day forecast?
4. Can you assess on computational complexity and runtime of the approach compared to the baselines?

**Details Of Ethics Concerns:**

--

---

> ### Author Response · Authors · 2023-11-20
> **Response to Comment 1 for Reviewer SkxH**
>
> Thank you very much for your careful reading and very insightful comments. We are grateful for your encouraging comments including calling our core idea "original" and "well-positioned to existing work", as well as our results "state-of-the-art." We have carefully reviewed all your comments and revised our manuscript. Due to space constraints, the new and revised contents have been integrated not only into the relevant sections of the main body, but also extensively into the appendix. We offer detailed responses to your comments below.
>
>
> **Comment 1: The reasons given to support the claim that univariate financial time series data (in general) embody recurring complex patterns over time appear vague and lack thorough evaluation. For example, the authors consider only a single financial data set (i.e., the S&P 500 index), and the analytical approach seems very fine-grained and not very general. It would be helpful to have insights into the clustering results, i.e., indicating the actual number of classes as well as the number of items per class, or showing multiple centers, etc. And even though Appendix A/B broadens some aspects there remains a lack of rigor in the arguments/results.**
>
> **Response:** Thank you for pushing us to provide strong justification for recurring complex patterns in financial time series data. This is a central claim in our paper, and the foundation for both our methodological contribution and empirical performance. We would like to lean on financial domain knowledge a bit here. Traders and investors have been analyzing recurring patterns (often referred to as "technical analysis") for over one hundred years [1], with colorful names such as "head-and-shoulders" and "upside down flag" for distinct shapes. The academic finance community has explored chaos theory, a branch of applied mathematics with a central focus on recurring complex patterns, to financial data [2,3,4]. These recurring patterns are not only confined to the S\&P 500 index, but also appear in other financial time series including individual stocks and futures contracts (we explore all three types of assets in our empirical work). In our experiments, we indeed identify recurring patterns within these financial assets. Notably, we uncover some of the same patterns (cluster centers) as those found in established technical analysis, as depicted in Appendix B.2. This consistency with long-standing domain knowledge provides support to the notion of recurring complex patterns within the data and demonstrates the effectiveness of our approach in identifying genuine patterns. For instance, our approach identifies 14 distinct patterns within S\&P 500 in our sample period, with each pattern corresponding to approximately 70 subsequences sliced from the entire time series on average.
>
> A major challenge in modeling the recurring complex patterns present in financial data is the automatic detection of these patterns without human input. Because the actual number and identity of the clusters in the real-world financial data remain unknown, we cannot directly evaluate the merit of our clustering algorithm on real data. To test the effectiveness of our approach, we have conducted a deeper investigation into the pattern recognition module using simulated time series data. These time series data comprise multiple scale-invariant patterns that we define, exhibiting varying lengths and magnitudes. The key advantage of these simulated data is that we have full control over the data-generating process. This control allows us to know the ground truth, against which we can compare and evaluate the model performance. Our investigation shows that the pattern recognition module effectively learns the correct ground-truth clusters. We have included the detailed settings and results of this investigation in the updated Appendix B.3 on page 14-16 of the revised manuscript.
>
> [1] Lo, Andrew W., Harry Mamaysky, and Jiang Wang. "Foundations of technical analysis: Computational algorithms, statistical inference, and empirical implementation." Journal of Finance. 2000.
>
> [2] Mantegna, Rosario N., and H. Eugene Stanley. "Scaling behaviour in the dynamics of an economic index." Nature. 1995.
>
> [3] Peters, Edgar E. "Fractal structure in the capital markets." Financial Analysts Journal. 1989.
>
> [4] Budinski-Petković, Lj, I. Lončarević, Z. M. Jakšić, and S. B. Vrhovac. "Fractal properties of financial markets." Physica A: Statistical Mechanics and its Applications. 2014.

---

> ### Author Response · Authors · 2023-11-20
> **Response to Comment 2 and 3 for Reviewer SkxH**
>
> **Comment 2: I lack understanding of the learning aspects of the pattern evolution network $\phi$. How do you extract the information used from the training data? I can't find any more detailed information about the Info() function in line 6 of Algorithm 2.**
>
> **Response:** Thank you for highlighting this potential ambiguity in the pattern evolution network. Indeed, the term "information" be somewhat misleading. In this context, "information" refers to the attributes of transition states, including pattern types, duration, and magnitudes.
>
> The role of the pattern recognition module is to extract these specific attributes related to the transition states of consecutive patterns. The pattern evolution network, denoted by $\phi$, is tasked with learning the Markov transition probabilities of these states between successive patterns. That is, given the current pattern and its states, $\phi$ is designed to predict the subsequent states.
>
> To improve the clarity of our exposition and eliminate any potential confusion, we have revised the description pertaining to this aspect of the model in the latest version of the manuscript.
>
> **Comment 3: Overall, I am not convinced that the proposed pattern-centric approach really adds complementary information to learning financial time series data. I suppose this is primarily due to deficiencies in the evaluation of the pattern recognition module. I also miss a detailed description of the included datasets and results reproducibility is very limited due no code submission.**
>
> **Response:** Thank you for helping us strengthen the core of our approach, which indeed depends on patterns. We would like to address your concern from two perspectives.
>
> First, we do believe that the pattern-centric methodology offers important and useful information that improves performance. We include an ablation study in Appendix E.4 which contains an FTS-Diffusion variant that uses a fixed-length segmentation. This setting is akin to turning off the pattern recognition module, such that the variant is pattern agnostic. We find that this variant results in diminished performance compared to the pattern-centric approach. This observation underscores the value of the pattern-centric approach for effective modeling of financial time series. We have tried to make these points clearer in the revised manuscript.
>
> Second, by focusing on patterns and segments, we have effectively decomposed the entire time series into its smaller, more manageable components. We can then effectively model these components, before assembling them back into a coherent series. This decomposition is likely to reduce the complexity of the generation task, as we transform the challenge of learning a complex distribution to more well-behaved, simpler patterns. Our additional investigations, as outlined in our response to Comment 1, further substantiate that our pattern recognition module can accurately decompose simulated time series that encompass a variety of scale-invariant patterns into respective segments that correspond to each pattern. This exercise offers additional evidence that this module successfully achieves its objective to break down the complicated distribution of the entire time series into its constituent parts.
>
> We apologize for the confusion regarding the included datasets. We provide a description of the dataset in Section 5.1 which contains the three time series, covering three types of assets (S\&P 500 index, Google's stock price, and Corn futures), as well as their date ranges that we used in our experiments. Per your comment, we have made them more visible with additional descriptions in the revised manuscript.
>
> We understand your concern about the reproducibility. As you are aware, research code is not necessarily always in a format amenable to immediate sharing. We will be glad to share the code upon the acceptance of the paper after we have had the chance to clean it up. Thank you for your understanding.

---

> ### Author Response · Authors · 2023-11-20
> **Response to Questoin 1-4 for Reviewer SkxH**
>
> **Question 1: I suppose the length of the entire time series is $T=\sum_{m=1}^M t_m - t_{m-1}$?**
>
> **Response:** Thank you for your clarifying question, which pushes us to double-check that our notation is consistent. The $m$-th segment, with the length of $t_m$, is represented as a vector $x_m$={$x_{m,1}$,...,$x_{m, t_m}$} in our problem statement. Please note that the index for the length begins at $1$ and concludes at $t_m$. Accordingly, the length of the entire time series is the sum of the lengths of all segments, expressed as $T=\sum_{m=1}^M t_m$.
>
> **Question 2: In Eq. (3), $\beta \leq 1$?**
>
> **Response:** The parameter $\beta$ regulates the range of magnitudes. As you point out, for financial data, $\beta \leq 1$ is quite reasonable as daily returns are extremely unlikely to change by more than 100\% (perhaps with the exception of some novel cryptocurrencies). For the types of assets we consider including a stock market index, a large-cap stock, and a commodity futures, we can further leverage domain knowledge regarding these assets and stipulate that $\beta \leq 0.2$ since it is exceedingly rare to observe a price change greater than 20\%.
>
> **Question 3: In case of a one-day ahead forecast I would have expected comparison to a very naive approach, e.g., using the value of the previous day as prediction; especially in reference to the Markov property. Can you additionally include a multiple-day forecast?**
>
> **Response:** Thanks for your suggestion to expand our empirical analysis. We have added the naive approach as a comparison, and we have conducted additional analysis on a five-day forecasting in the updated Appendix E.3.
>
> **Question 4: Can you assess on computational complexity and runtime of the approach compared to the baselines?**
>
> **Response:** We have outlined the computational complexity and runtime of our approach versus the baselines in the additional Appendix E.6. Due to its multi-module nature, our FTS-Diffusion has a higher complexity. This represents a trade-off between high fidelity and complexity. Our model achieves superior generation quality at the expense of somewhat increased complexity. In future work, we would like to explore ways to reduce the complexity of our method without sacrificing too much performance. Thank you for this useful suggestion.

---

> > ### Comment · Reviewer_SkxH · 2023-11-21
> >
> > I thank the authors very much for their thorough response and for the detailed discussion
> > of my questions and concerns.
> >
> > (add reproducibility)
> > The provision of code at the time of publication is satisfactory from my point of view, thank you.
> >
> > (add Question 1)
> > I confused $T$ with the highest possible point in time reached in a given time series; I apologize.
> >
> > You have clarified things, and I will increase my score correspondingly, under the assumption that these clarifications will make it into the updated version of the paper.
> >
> > Thank you again.

---

> > > ### Author Response · Authors · 2023-11-21
> > > **Response to the Further Comment for Reviewer SkxH**
> > >
> > > Thank you very much for approving our responses and raising the score. Your constructive comments have greatly helped us to enhance our manuscript. We appreciate your insightful input.

---

### Official Review · Reviewer_XZvW · 2023-11-01

**Soundness:** 2 fair
**Presentation:** 3 good
**Contribution:** 3 good
**Rating:** 8
**Confidence:** 3

**Summary:**

This paper proposes the FTS-Diffusion framework for capturing irregular and scale-invariant patterns in financial data. The framework consists of three parts: pattern recognition, pattern dependent generation and pattern evolution prediction. Experiments show that the data generated by this framework outperforms existing generation methods in terms of distribution matching and downstream task performance.

**Strengths:**

1. The irregular and scale-invariant characteristics of stock data compared with other time series data are explained clearly, and it is a relatively novel perspective.
2. The design description of the three modules of FTS-Diffusion is clear and reasonable.
3. The experimental results exhibits promising performance of synthetic data generation with FTS-Diffusion in terms of distribution matching with real data and downstream prediction performance.

**Weaknesses:**

1. The experiments are only conducted on  several types of stock data and did not include more scenarios in the financial field.
2. Lack of ablation experiments. Three modules, and the effectiveness of some detailed designs in each module have not been effectively explained
3. No rolling window experiment was performed, the results on a single train/test split are not reliable enough, and some detailed experimental settings (hyper-parameters, train/test split details) are not explained

**Questions:**

1. Whether the pattern combination found by FTS-Diffusion has good generalization to overcome the serious distribution shift problem in financial scenarios
2. Can FTS-Diffusion be used for direct prediction?

---

> ### Author Response · Authors · 2023-11-20
> **Response to Comment 1, 2, and 3 for Reviewer XZvW**
>
> Thanks for your insightful feedback. We are encouraged by your comments that the irregular and scale-invariant characteristics in stock data is "relatively novel perspective", our experimental results are "promising", and our three-module design is "clear and reasonable". We have taken your feedback into careful consideration and made corresponding revisions to our manuscript. Due to space constraints, the updated contents have been incorporated not only into the relevant sections of the main body, but also extensively into the appendix. We provide a detailed response to your comments below.
>
> **Comment 1: The experiments are only conducted on several types of stock data and did not include more scenarios in the financial field.**
>
> **Response:** We take your point that a broad set of empirical experiments using different types of financial time series data would offer more convincing evidence in support of our proposed methodology. Indeed, any method that claims to capture the data-generating process of financial data should be applied to a variety of series. To this end, we include three types of assets in our analysis: A broad stock market index (S\&P 500 Index), an individual stock (Alphabet, ticker GOOG), and a commodity futures contract (Corn traded on the Chicago Board of Trade). We view these three series as capturing disparate assets - a broadly-diversified stock portfolio behaves differently from a single stock or from a financial derivative. By including three types of series with varying characteristics, we hope to showcase the versatility of our methodology.
>
> To the best of our knowledge, this is the first work to systematically model and generate the sought-after irregular and scale-invariant patterns in financial time series. Given that there are already much to investigate as in such a first step and much to cover in this 9-page article, we found it difficult to include more assets. We are certainly interested in applying our approach to other time series, which is an important and interesting next step.
>
> **Comment 2: Lack of ablation experiments. Three modules, and the effectiveness of some detailed designs in each module have not been effectively explained.**
>
> **Response:** Thank you for this comment. We clarify that we agree that ablation experiments are important in any newly proposed method, and we have already included an ablation study in Appendix E.4 in the initial submission. In the study, we investigate the contribution of each module in our FTS-Diffusion, by conducting experiments on variants of our model, removing one module each time. In our revised manuscript uploaded, we have enhanced the clarity of the relevant discussions in the main text to avoid potential future confusion; please refer to the updated Appendix E.4.
>
> **Comment 3: No rolling window experiment was performed, the results on a single train/test split are not reliable enough. Some detailed experimental settings (hyper-parameters, train/test split details) are not explained.**
>
> **Response:** We appreciate your reminder that a single train/test split is potentially not sufficiently reliable to claim the superiority of our approach. In our experiments, we employ an 80/20 split, using the first 80\% for training and the remaining 20\% for testing. We recognize that this single, static split is not enough, and we follow your recommendation to include the results of additional experiments employing various train/test splits; please refer to the additional Appendix E.5 on page 20 in the updated manuscript. We also include rolling window experiments, adhering to common practice in the finance literature. We find that the takeaway from these experiments is consistent with our original experiment, which affirms the robustness of our approach.
>
> We have attached and explained the key hyper-parameters for each module in their corresponding Appendices B.1, C.2, and D.1, respectively. In particular, within the pattern recognition module, we set the number of clusters and range of segment lengths by leveraging domain knowledge. For the generation and evolution modules, we build the networks following the typical settings, including the number of layers and the diffusion steps, in the relevant literature and train them using the Adam optimizer with adaptive learning rates. In our experiments, we employed an 80/20 split strategy for our datasets, using the first 80\% as the training set and the remaining 20\% as the test set; please refer to the relevant description in Section 5.1 on page 7 in the updated manuscript.

---

> ### Author Response · Authors · 2023-11-20
> **Response to Question 1 and 2 for Reviewer XZvW**
>
> **Question 1: Whether the pattern combination found by FTS-Diffusion has good generalization to overcome the serious distribution shift problem in financial scenarios?**
>
> **Response:** Thanks for your insightful question. Indeed, distribution shifts in financial time series present significant challenges for any model. The return distribution (or even the data-generating process) in November 2019, when the economy was humming along, is potentially different from that of March 2020, when Covid-19 affected our social behavior as well as the real economy. A shift in the return distribution manifests in two possible ways in our framework. First, the set of possible patterns may change. Perhaps in November 2019, there are 15 unique patterns that make up the distribution, whereas there are 19 patterns in March 2020. Second, the transition probabilities that govern the dynamic evolution of patterns, magnitudes, and intervals might be different under one regime versus another. For example, it is well known that volatility tends to cluster in financial returns, such that large changes in prices are followed by more large changes, while the persistence of this dynamic may differ across regimes. Our model will capture a change in persistence through its transition parameters controlling consecutive magnitude values.
>
> In handling the two above possibilities, our approach is robust to the distribution shifts concerning the number of patterns as our pattern identification module is trained over the full range of market conditions throughout the entire history. This way, as long as our training data covers different market conditions, we are likely to recover the full set of possible patterns. However, the model is not well-suited to deal with shifts in transition probabilities. We model transition probabilities under a Markov assumption that applies fixed probability across the entire time series. A potential solution to this challenge would be replacing the latent process of our pattern evolution network with Markov-switching modeling to allow for time-varying transitions. As the first work on the topic, our focus in this paper is on identifying irregular, scale-invariant patterns in financial data - a non-trivial problem as it stands - we leave the treatment of the distribution shift problem for future work, and we have made it clear in the updated conclusion section on page 9 in the uploaded manuscript.
>
> **Question 2: Can FTS-Diffusion be used for direct prediction?**
>
> **Response:** Thanks for your perceptive question, which compelled us to think harder about whether we really need a separate prediction model, or if prediction can be done in one step through the generative diffusion model. Our FTS-Diffusion is specifically designed to generate synthetic financial time series based on irregular and scale-invariant patterns. In this capacity, FTS-Diffusion can generate synthetic data that assists in, and potentially improves, the predictive power of other models. While FTS-Diffusion appears capable of capturing the inherent structure of financial time series data, it is not optimized for prediction. As such, its direct predictive ability is likely to be quite limited. It would be interesting to extend the generative model to include predictive capabilities.

---

> ### Author Response · Authors · 2023-11-22
> **Gentle Reminder**
>
> Dear Reviewer XZvW,
>
> We hope our responses find you well and address your concerns regarding our work. We kindly wanted to check if you have any further questions or need additional clarification.
>
> We value your insights and stand ready to make any necessary revisions based on your feedback. Thank you once again for your time and help.
>
> Best,
>
> Authors

---

> > ### Comment · Reviewer_XZvW · 2023-11-23
> >
> > Thank you for your detailed responses, which answer most of my concerns. I will raise my score.

---

> > > ### Author Response · Authors · 2023-11-23
> > > **Response to Further Comment for Reviewer XZvW**
> > >
> > > Thank you for acknowledging our responses and raising the score. We greatly appreciate your comments that have been instrumental in refining our manuscript. Thanks for your valuable feedback again.

---

### Official Review · Reviewer_dpJk · 2023-11-06

**Soundness:** 2 fair
**Presentation:** 3 good
**Contribution:** 3 good
**Rating:** 6
**Confidence:** 2

**Summary:**

The paper proposes a new generative framework called FTS-Diffusion for synthesizing financial time series data that exhibit irregular and scale-invariant patterns. The framework consists of three modules: Pattern recognition module, Pattern generation module and Pattern evolution module. The paper demonstrates that FTS-Diffusion can generate synthetic financial series that closely match the distribution and properties of real data. The synthetic data also improves performance when used for downstream prediction tasks. The proposed framework is the first capable of generating complex time series with irregular and scale-invariant patterns. It addresses the problem of data shortage and improves reliability of deep learning models in financial applications.

**Strengths:**

Originality: The paper proposes a novel 3-module framework specifically designed to model irregular and scale-invariant patterns. The scale-invariant pattern recognition algorithm SISC is also a novel technique.

Quality: The paper is well-written and easy to follow. However, I have some concerns regarding the experiment part, which I will discuss in details in the "Weaknesses" section.

Clarity: The problem is clearly motivated through examples and definitions. The method description provides sufficient details to understand the key designs. The writing clearly conveys the core ideas and contributions.

Significance: The work addresses an important problem in financial time series modeling and generation. It has high practical significance for improving reliability of deep learning in finance under data shortage. The framework could potentially generalize to other irregular and scale-invariant time series beyond finance.

**Weaknesses:**

1. The main weakness of this paper is its experiment part. Specifically, the data / training details are not clear, but it's very important in financial time series forecasting as it's very easy to have "label leakage". Is it possible to include more data / training details? In particular, what data are used to train each module / model, what's data splitting strategy, what data are used for testing?
2. It would be better to include some standard metrics like Sharpe Ratio in the experiment section.

I'd like to raise my score once my concerns regarding experiments are addressed.

**Questions:**

1. It seems the current experiments only use price returns as input. It's standard in industry to use a lot more derived features from price data / order book data, so I am wondering whether it's easy to incorporate other features into the current framework.
2. In Figure 6(a), it's very surprising that regardless of the "synthetic proportion", the performance is about the same, meaning that the synthetic data is almost as good as the original data, which is very surprising. Do we have any hypothesis on why the synthetic data can be as good as the original data?
3. If it's possible, it would be good to include code in the "Supplementary Material" for better reproducibility.

---

> ### Author Response · Authors · 2023-11-20
> **Response to Comment 1 and 2 by Reviewer dpJk**
>
> Thank you for your constructive comments, and your encouragement in pointing out the originality of our approach ("SISC is also a novel technique") and clearly identifying our problem statement and motivation ("the work addresses an important problem in financial time series modeling and generation and has high practical significance for improving the reliability of deep learning in finance under data shortage"). We have carefully considered your comments and revised our manuscript accordingly. Due to the space limitation, we have included the new and revised contents in not only the corresponding sections in the main body but also substantially in the appendix. Please find our responses below.
>
> **Comment 1: The data/training details are not clear, but it's very important in financial time series forecasting as it's very easy to have"label leakage". In particular, what data are used to train each module/model, what's data splitting strategy, what data are used for testing?**
>
> **Response:** Thank you for pointing out the importance of the experimental settings and the potential "label leakage" issue. We take this point to mean that in testing the performance of a predictive model, it is possible that the model obtains certain information about the outcome variable if we are not careful in separating the test and training sets. In our experiments, we employ an 80/20 split, using the first 80\% for training and the remaining 20\% for testing. FTS-Diffusion is trained on the first 80\% of the data, and so is the downstream LSTM prediction model. To be clear, suppose the first 80\% of the data contain data samples {$T_{train}$} = {$t_1$, ..., $t_i$}, whereas the last 20\% contain data samples {$T_{test}$} = {$t_{i+1}$, ..., $t_n$} where $n$ is the length of our entire data. All three modules of FTS-Diffusion - pattern recognition, generation, and evolution - are trained only using {$T_{train}$}. In the downstream prediction model, the LSTM-based prediction model is also trained on {$T_{train}$}. Therefore, neither the generation model nor the downstream prediction model has seen any of the test set samples {$T_{test}$} in the training phase. For the experiments in which we create synthetic data to be used by the downstream model, the generative model still has only seen {$T_{train}$}. In the testing phase, entirely new data are presented to the prediction model. Therefore, we believe that our approach prevents the possibility of label leakage. We have tried to make the description of our training and testing setting clearer in Section 5.1 on page 7 of the revised manuscript.
>
> **Comment 2: It would be better to include some standard metrics like Sharpe Ratio in the experiment section.**
>
> **Response:** Thanks for your suggestion. In our experiments, we want the synthesized time series to resemble actual data both statistically and economically. The stylized facts (e.g., heavy tails) and distribution tests (such as the Kolmogorov-Smirnov test of identical distributions) in our experiments serve to illustrate the statistical similarity between the synthetic and observed time series, adhering to common metrics used in the financial time series generation literature (for example, by J.P. Morgan AI Research [1]). We understand your comment as urging us to add further economic comparisons. We follow your suggestion and have incorporated the Sharpe ratio, Sortino ratio, and Calmar ratio; please refer to the revised Appendix E.2. These summary statistics are core characteristics of financial asset returns, and different asset classes such as stocks and bonds tend to have quite different values. The synthetic data generated by FTS-Diffusion exhibit comparable statistics to those of observed data, which provides further evidence supporting the effectiveness of our approach in generating realistic financial time series.
>
> [1] Assefa, Samuel A., Danial Dervovic, Mahmoud Mahfouz, Robert E. Tillman, Prashant Reddy, and Manuela Veloso. "Generating synthetic data in finance: opportunities, challenges and pitfalls." Proceedings of the ACM International Conference on AI in Finance. 2020.

---

> ### Author Response · Authors · 2023-11-20
> **Response to Question 1, 2, and 3 for Reviewer dpJk**
>
> **Question 1: It seems the current experiments only use price returns as input. It's standard in industry to use a lot more derived features from price data/order book data, so I am wondering whether it's easy to incorporate other features into the current framework.**
>
> **Response:** Thank you for your insightful question regarding the order book. Order book information is most important for intraday prediction, where market microstructure dominates. Our empirical work focused on daily returns is less likely to be affected by order book dynamics. That being said, we understand the broader scope of the question as to whether it is feasible to integrate additional inputs, beyond price data, into the prediction model in our downstream experiment in Section 5.3. The LSTM-based network can accommodate additional features beyond prices. For example, [2] illustrates that an LSTM prediction model can incorporate news event and media sentiment, in addition to prices, in predicting stock movements. We choose to only use prices as inputs to keep the downstream model as simple as possible while illustrating the usefulness of our generative model.
>
> [2] Li, Qing, Jinghua Tan, Jun Wang, and Hsinchun Chen. "A multimodal event-driven LSTM model for stock prediction using online news." IEEE Transactions on Knowledge and Data Engineering (TKDE). 2020.
>
> **Question 2: In Figure 6(a), the synthetic data is almost as good as the original data, which is very surprising. Do we have any hypothesis on why the synthetic data can be as good as the original data?**
>
> **Response:** Thank you for pointing this out. The experimental results in Figure 6(a) are indeed encouraging, as they suggest that our FTS-Diffusion effectively captures the most relevant structure and dependencies, which manifest as recurring patterns, within the observed data. In other words, the scale-invariant, recurring patterns that we identify summarize the most important variation in financial time series data, at least in our setting. As a result, the generated synthetic time series stands comparably with the observed data, particularly from the perspective of the downstream prediction model. In our future investigation, we will carry out more extensive experiments to further evaluate the performance of our approach under various real-world settings.
>
> **Question 3: It would be good to include code in the "Supplementary Material" for better reproducibility.**
>
> **Response:** Thanks for your suggestion. We will be glad to share the code upon the acceptance of the paper, after we have had the chance to clean it up, as is common practice. Thank you for your understanding.

---

> > ### Comment · Reviewer_dpJk · 2023-11-20
> >
> > I updated my score from 5 to 6, thanks for the great rebuttal.
> >
> > One more qq:
> > 1. how did you tune the parameters, since you only have training dataset and test dataset (but no validation dataset)?

---

> > > ### Author Response · Authors · 2023-11-20
> > > **Response to Further Comment for Reviewer dpJk**
> > >
> > > Thank you for acknowledging our responses and raising the score. We deeply appreciate your feedback to improve the clarity and quality of our work.
> > >
> > > In response to your further question about tuning our model, we approach it from two perspectives: the method we used for parameter tuning and the need for a validation set in our case.
> > >
> > > Initially, we start with a set of key hyper-parameters that are known to generally work well, as indicated in the literature. We then tune the parameters by monitoring the training loss and visually inspecting the quality of the generated samples during training. This gives us a qualitative overview of how well the model learns the underlying data distribution.
> > >
> > > Regarding the validation set, deep generative learning does not have a clear target "label" to compare against during training, making the validation less straightforward in common practice for the same purposes as in supervised learning. In particular, our primary focus lies in the training of our financial time series generation model, while the downstream prediction model serves to evaluate whether augmenting the dataset with synthetic data generated by our model can improve its overall performance. Consequently, we maximize the use of the available data for the training of our generation model, rather than reserving a portion for a separate validation set.

---

### Author Response · Authors · 2023-11-23
**General Comment for All Reviewers**

Dear Reviewers,

Thank you once again for your helpful comments. We are pleased to know that our responses have addressed your concerns.

In the latest revision, we've taken additional steps to enhance the clarity and readability of the figures in our manuscript.

Best regards,

Authors

---

### Meta-Review · Area_Chair_uCBn · 2023-12-13

**Metareview:**

This paper is concerned with the problem of generative learning from financial time series data, for which limited data availability poses a big challenge.  It proposes FTS-Diffusion, a generative framework to select features and model intrinsic patterns of univariate financial time series data with the aim of overcoming data shortage. The model consists of three building blocks: recognition of irregular and scale-invariant patterns in each time series sample, generation of synthesized segments of patterns, and evolution of the latter over time. Extensive experiments show that FTS-Diffusion generates synthetic financial time series highly resembling observed data, outperforming state-of-the-art alternatives.

The proposed framework seems novel, and the paper is well written. The experimental results on synthetic data generation look encouraging. On the other hand, the work will benefit from experiments in more diverse environments, necessary ablation studies, and rolling-window experiments.

**Justification For Why Not Higher Score:**

The work will benefit from experiments in more diverse environments, necessary ablation studies, and rolling-window experiments.

**Justification For Why Not Lower Score:**

The proposed framework seems novel, and the paper is well written. The experimental results on synthetic data generation look encouraging.

---

### Decision · Program_Chairs · 2024-01-16

Accept (spotlight)